# Graph World Model

Tao Feng [1]  Yexin Wu [1]  Guanyu Lin [1]  Jiaxuan You [1]

## Abstract

World models (WMs) demonstrate strong capabilities in prediction, generation, and planning tasks. Existing WMs primarily focus on unstructured data while cannot leverage the ubiquitous structured data, often represented as graphs, in the digital world. While multiple graph foundation models have been proposed, they focus on graph learning tasks and cannot extend to diverse multi-modal data and interdisciplinary tasks. To address these challenges, we propose the Graph World Model (GWM), a world model that supports both unstructured and graph-structured states with multi-modal information and represents diverse tasks as actions. The core of a GWM is a generic message-passing algorithm to aggregate structured information, either over a unified multi-modal token space by converting multi-modal data into text (GWM-T) or a unified multi-modal embedding space by modality-specific encoders (GWM-E). Notably, GWM introduces action nodes to support diverse tasks, where action nodes are linked to other nodes via direct reference or similarity computation. Extensive experiments on 6 tasks from diverse domains, including multi-modal generation and matching, recommendation, graph prediction, multi-agent, retrieval-augmented generation, and planning and optimization, show that the same GWM outperforms or matches domain-specific baselines' performance, benefits from multi-hop structures, and demonstrate strong zero-shot/few-shot capabilities on unseen new tasks. Our codes for GWM is released at https://github.com/ulab-uiuc/GWM.

[1]Department of Computer Science, University of Illinois Urbana Champaign Urbana, IL, USA. Correspondence to: Tao Feng <taofeng2@illinois.edu>, Jiaxuan You <jiaxuan@illinois.edu>.

*Proceedings of the 42nd International Conference on Machine Learning*, Vancouver, Canada. PMLR 267, 2025. Copyright 2025 by the author(s).

## 1. Introduction

A world model (WM) (Ha & Schmidhuber, 2018) constructs the world observations as states and predicts future states based on given actions. Modern world models are trained with massive data (Liu et al., 2024b; Cui & Gao), demonstrating successful prediction, generation, and planning capabilities. However, existing world models do not directly generalize to structured data, primarily graphs, that are ubiquitous in science (Jin et al., 2018; You et al., 2018) and industry (Ying et al., 2018; You et al., 2022) and can be further enriched with multi-modal information (Ektefaie et al., 2023). Therefore, our paper aims to raise attention to this pressing research question: *Can we extend a WM to handle graph-structured data across a broad range of tasks?*

Existing WMs mainly focus on unstructured data. For example, iVideoGPT (Wu et al., 2024a) and Genie (Bruce et al., 2024) are successful world models over video data. However, the relations and structures in the data are rarely explored in these works. Although some works (Zhang et al., 2021; Zhu et al., 2022) attempt to model structured data in WM using graphs, they have focused solely on planning problems in a specific domain. In recent years, researchers have also explored the concept of the Graph Foundation Model (GFM) (Liu et al., 2023a; Chen et al., 2024a). However, these methods are confined to predefined graph learning tasks, which cannot easily extend to: (1) multi-modal input data including images and text, (2) diverse tasks beyond standard graph prediction tasks, and (3) data without explicit structure, *i.e.*, standard unstructured data.

To address these challenges, we propose the Graph World Model (GWM) that embeds the capabilities of the graph into the WM, which models the current state as a graph and the action as a node (see Table 1 for comparison with existing methods). Various tasks can be expressed as action nodes; for example, in a graph prediction task, predicting the label of a given node/edge/subgraph leads to *intended* action nodes that link relevant nodes in the state graph, *i.e.*, target nodes, to the action node; in a retrieval-augmented generation (RAG) task, we can also represent a user query as an *unintended* action node that links to target nodes in the state graph via embedding similarities.

To build GWMs, we first introduce a simplified token-based GWM (GWM-T), which integrates multi-modal data like

*Table 1.* **Comparison with existing representative works from three perspectives: task type, data structure, and model type.** Compared with existing WM and GFM, GWM can tackle multi-domain tasks and be applied to both structured and unstructured data.

| Method | Task Type | Data Structure | Model Type |
|---|---|---|---|
| Genie (Bruce et al., 2024) | Video generation | Unstructured | WM |
| $L^3P$ (Zhang et al., 2021) | Planning | Structured | WM |
| BioBridge (Wang et al.) | Biomedical domain | Structured | GFM |
| LLAGA (Chen et al., 2024a) | Graph domain | Structured | GFM |
| GWM-T | Multiple domains | Both | GWM |
| GWM-E | Multiple domains | Both | GWM |

image, table, and text into text modality and represents them as nodes in a graph state. We further develop a token-level message-passing algorithm that aggregates the neighbor information to update the text representation of the state node. Finally, the target nodes on the state graph and prompted action nodes will be fed into multi-modal decoders such as LLMs and Stable Diffusion (Rombach et al., 2022). Despite its simplicity, GWM-T sometimes suffers from high token costs and limited context length. Inspired by latent diffusion models, which introduced modeling in latent space rather than directly on pixels like diffusion to enhance model performance and efficiency, we further develop an embedding-based GWM (GWM-E). GWM-E first employs modality-specific encoders to process different modalities into node embeddings. Then it utilizes embedding-level message passing to update the node embedding. Finally, the multi-modal information in target state nodes is consolidated through a multi-hop projector before passing them to the decoders.

We conduct extensive experiments on 6 tasks from diverse domains, including world prediction (multi-modal generation and matching, recommendation, graph prediction), world generation (multi-agent collaboration, retrieval-augmented generation), and world optimization (planning and optimization), with both proposed GWM variants and domain-specific baselines. Results show that (1) GWMs generalize across domains, as the same GWM outperforms or matches domain-specific baselines' performance, (2) graph information matters in GWM, as GWMs benefit from multi-hop graph information, and (3) GWMs demonstrate strong zero-shot/few-shot capabilities on unseen new tasks.

## 2. Graph World Model

### 2.1. World Model Preliminaries

A world model aims to predict future states based on the current state and action, which contains the following main components: **(1) State**. The state $s$ of the world model estimates the observation of the world. It usually consists of multi-modal information and the state of the $t$ step/time slot can be depicted as $s_t$. **(2) Action**. The action $a_t$ at step/time

$t$ is task-related. It can be a real-world operation, a code function in the digital world, and even some queries and instructions. **(3) Transition**. The transition $P(s_{t+1}|s_t, a_t)$ depicts the transit probability from the current state $s_t$ to its next state $s_{t+1}$ after the execution of action $a_t$.

### 2.2. Multi-modal World Represented by Graphs

**Graph for state modeling**. We define the world state as a graph $\mathcal{G} = (\mathcal{V}, \mathcal{E})$ to represent multi-modal data with complex relationships, shown in Figure 1. Specifically, $\mathcal{V} = \{v\}$ represents the set of nodes where each node $v = [v^a, v^b, v^e]$ consists of multi-modal information, including image $v^a$, table $v^b$, and text $v^e$ modality; when a modality is absent, the corresponding tensor will be empty. $\mathcal{E} = \mathcal{E}_p \cup \mathcal{E}_m\}$ is the edge set consists of explicit edges $\mathcal{E}_p$ and implicit edges $\mathcal{E}_m$. Explicit edges $\mathcal{E}_p$ are often those established through expert knowledge or ground truth observations. For example, in the ogbn-arxiv dataset (Hu et al., 2020), edges are determined based on references between papers and historical collaborations between authors. Implicit edges $\mathcal{E}_m$ are those constructed through connections represented by embedding similarities in the dataset. A typical example is in many protein datasets (Heumos et al., 2023; Stuart et al., 2019; Stuart & Satija, 2019), where edges are obtained based on the similarity of certain node feature embeddings.

**Different levels of world action and state transition**. We model the action $a$ as an action node that queries the current state nodes $v$ to obtain the target nodes $v_r$ using function $R$, whose process can be formulated as $v_r = R(v, a)$. We further categorize the world's actions into two types, as shown in Figure 1: one is directly related to the specific structures on the graph, called intended action $a_d$. It includes three levels: node-level, edge-level, and graph-level. The other is indirectly related to the specific structures on the graph through semantic relations such as Retrieval-augmented Generation (RAG), called unintended action $a_u$. As shown in Figure 1, to implement this action, we can first calculate the similarity between the action node and state nodes, and then retrieve the top-k state nodes for querying. According to the introduction in Section 2.1, we can conclude that an action causes a transition of state $s_{t+1} = f_{tr}(s_t, a_t)$, which includes three types: update nodes, update edges, and update graphs. Here, $f_{tr}$ means a transition function, which can be a neural network.

### 2.3. Instantiations of GWM

As shown in Figure 2, We have listed some representative instantiations that can be unified into a graph world model from three aspects: (a) world prediction, (b) world generation, and (c) world optimization.

**World prediction**. **(1) Multi-modal generation and matching.** As shown in Figure 2(a), the task includes

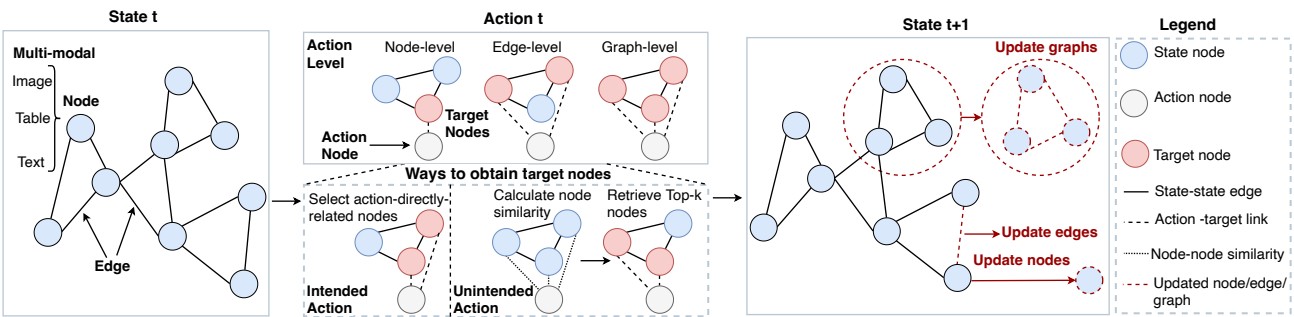

*Figure 1.* **Multi-modal world state transition can be modeled via graphs**. We model the current state as a graph and each node contains one or more modalities from image, table, and text. Further, the world action is modeled as an action node that queries the current state nodes. We categorize actions into two types: intended actions, which include three levels—node, edge, and graph—and unintended actions, whose implementation involves similarity computation similar to RAG. Finally, the transition function updates states at three different levels based on state and action: update nodes, update edges, and update graphs.

two subtasks. The multi-modal generation task (Rombach et al., 2022; Zhang et al., 2023a) involves predicting missing modalities given the available modal information and their interconnections. Specifically, it considers clusters of corresponding modalities (such as an image, table, and text describing the same entity) as state nodes, and the relationships between clusters, such as similarity, are treated as edges. Thus, the action node here is at the node level. The multi-modal matching task (Rombach et al., 2022), similar to CLIP's pre-training task (Radford et al., 2021), predicts the correspondence between modalities. It treats each modality as a state node and the correspondences between modalities (including cross-modality similarity relationships) (Jin et al., 2024) as edges. Here, the action node is at the edge level. **(2) Recommendation.** Recommendations (Ni et al., 2023; Isinkaye et al., 2015; Ko et al., 2022) are based on the historical interactions and features of users and items to predict future interactions, as shown in Figure 2(b). Specifically, it models the user nodes and item nodes as state nodes. Moreover, the action node is edge-level. **(3) Traditional graph prediction.** Traditional graph prediction (Kipf & Welling, 2016; Veličković et al., 2017b; Hamilton et al., 2017b) primarily focuses on three types of tasks: node-level, edge-level, and graph-level. We follow previous work's settings of nodes and edges and define the action nodes of three levels.

**World generation**. **(4) Multi-agent collaboration.** As shown in Figure 2(d), the purpose (Zhuge et al., 2024; Liu et al., 2023b; Wu et al., 2024b) of this task is to generate task-oriented outputs based on the interaction between agents and external knowledge, as well as communication among agents. Specifically, its state nodes consist of agent nodes with different profiles, along with multi-modal nodes in external knowledge. Its edges include agent-agent and agent-knowledge relationships. The action node is graph level and the target nodes primarily include various agent nodes (Zhuge et al., 2024; Liu et al.,

2023b). **(5) Retrieval-augmented generation.** The purpose of Retrieval-Augmented Generation (RAG) is to enhance the generation capabilities of Large Language Models (LLMs) by retrieving information from external knowledge (Lewis et al., 2020; Gao et al., 2023; Zhao et al., 2024). Recent studies such as GraphRAG (Edge et al., 2024; Peng et al., 2024) have shown that modeling the relationships between data chunks in external knowledge can enhance the generative capabilities of RAG. As illustrated in Figure 2(e), we model data chunks as nodes and the similarity of embeddings between chunks as edges. As introduced in Section 2.2, we design an unintended action node for RAG tasks.

**World optimization**. **(6) Planning and optimization.** World optimization involves generating the next best decision based on a sequence of historical decisions (Chen et al., 2021; Zheng et al., 2022; Siebenborn et al., 2022). Many studies have shown that modeling the relationships between historical decisions using graphs can enhance the decision-making effectiveness of world optimization (Jiang et al., 2018; Munikoti et al., 2023). Following them, we model decision states as nodes. The edges between these state nodes are often modeled based on their relationships, such as distance relationships (Prates et al., 2019) and the similarity of embeddings (Munikoti et al., 2023; Jiang et al., 2018). We model the action node as the graph level.

## 3. Token-based GFM

### 3.1. Multi-modality as token

One of the easiest ways to unify multi-modalities is to transfer them into text. Specifically, as shown in Figure 3, for image nodes $v^a$, we utilize a pretrained image-to-text LLaVA model (Liu et al., 2024a) $L$ to transform them into text nodes $v^{ta} = L(v^a)$. For table nodes $v^b$, we employ a table-prompt model $T$ to transform them into text nodes $v^{tb} = T(v^b)$ given the column names and feature values. Specifically, each value is paired with the corresponding column in the

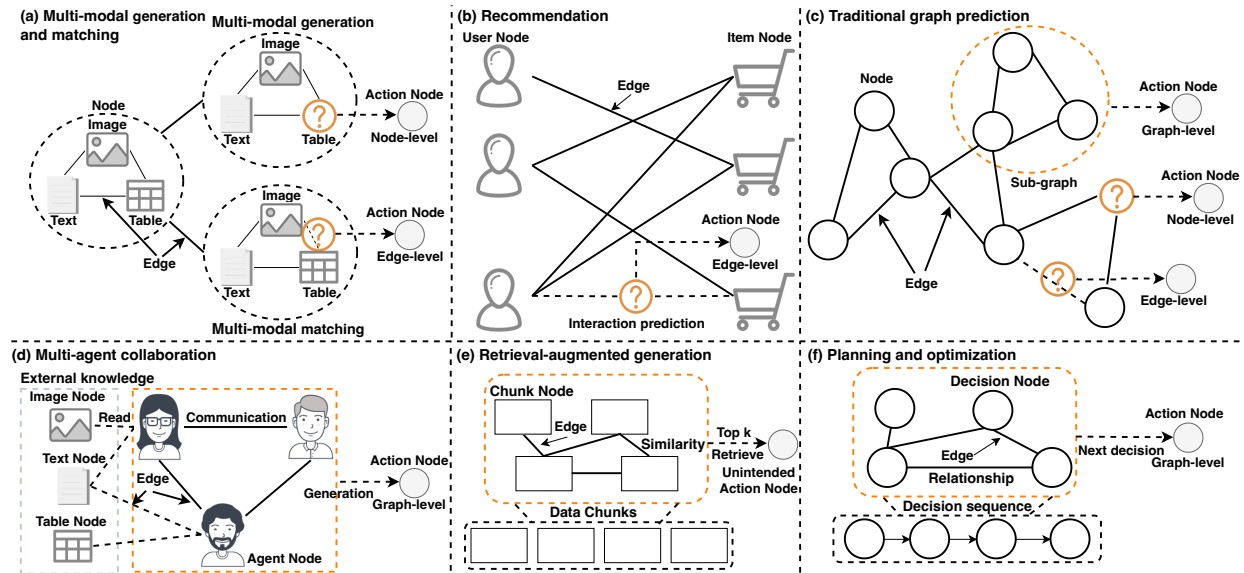

*Figure 2.* **Instantiations of GWM.** (a) Multi-modal generation and matching contains two sub-tasks. For multi-modal generation, it models modal clusters as nodes, whereas for multi-modal matching, it models nodes for each modality. It includes edges that represent inter-modal correspondences and cross-modal similarities. For these two types of subtasks, there are node-level and edge-level action nodes, respectively. (b) The state nodes of recommendation include user nodes and item nodes. Moreover, its edges are primarily derived from user-item interactions. We model edge-level action nodes to perform interaction prediction. (c) In traditional graph prediction, we follow existing work to construct task nodes and edges, and model three levels of action nodes according to different types of tasks. (d) The state nodes of multi-agent collaboration include agent nodes and multi-modal nodes from external knowledge. Its edges primarily consist of communications between agents and interactions between agents and external knowledge. We set up a graph-level action node to generate content based on the interactions of agents. (e) Retrieval-augmented generation treats each data chunk as a state node and builds edges through the embedding similarity between chunks. For this task, we have established unintended action nodes. (f) For planning and optimization, we model each decision as a state node and construct edges based on their relationships. We have established graph-level action nodes to generate the next decision.

format of "{column 1} is {value 1}, {column 2} is {value 2}, ...". Finally, we used a prompt template $P_u$ (specified Table 23 in Appendix C) to unify the three modal nodes into a single text node $v_c = P_u(v^{ta}, v^{tb}, v^e)$.

### 3.2. Token-level message passing

In contrast to traditional graph message passing (Kipf & Welling, 2016; Hamilton et al., 2017b; Veličković et al., 2017b), we employ token-level message passing here, which aggregates the text information of neighboring nodes. Specifically, as shown in the middle part of Figure 3, for the each unified text node $v_c$, the node embeddings update of the $l$-th layer is represented as:

$$\mathbf{h}_v^{(l)} = f_v\Big(\text{CONCAT}(\mathbf{h}_v^{(l-1)}, \{\mathbf{h}_u^{(l-1)}, u \in N(v)\})\Big), \quad (1)$$

where $\mathbf{h}_v^{(l)}$ is the node text presentation after $l$ iterations, $\mathbf{h_v}^{(0)}$ has been initialized as $\mathbf{h_v}^{(0)} = v_c$. In addition, $N(v)$ denotes the direct neighbors of node $v$ and $f_v(\cdot)$ denotes prompting strategy functions (specified in Table 24 of Appendix C) to unify nodes information of different hops.

### 3.3. Instruction tuning

Based on token-level message passing, we can obtain node text representations $\mathbf{h}_v$ for each node. Combining the discussion in Section 2.2, we identify the target nodes $v_r$ and their node text representations $\mathbf{h}_{vr}$, along with the action node $a$ and the state node. Further, we describe the action node using text and utilize a task-oriented prompt template $P_{sa}$ (specified in Appendix C) to combine the information from the target nodes and the action node, as shown in the right part of Figure 3. In response to the different modalities in next states, we designed two types of decoders. We first designed stable diffusion (SD) to generate images.

**Instruction tuning of SD**. SD operates by performing diffusion in a compressed latent space rather than directly on pixels. Initially, the system maps an input image $x$ to a lower-dimensional latent code $\mathbf{z} = \text{Enc}(x)$ through an encoder network. The generated latent representation $\mathbf{z}'$ is subsequently transformed back into image space via a decoder network, producing the final output $x' = \text{Dec}(\mathbf{z}')$. This latent representation $\mathbf{z}'$ is generated by the diffusion model using textual guidance from a prompt $c_T = P_{sa}(\mathbf{h}_{vr}, a)$. The fundamental optimization objective for training SD can

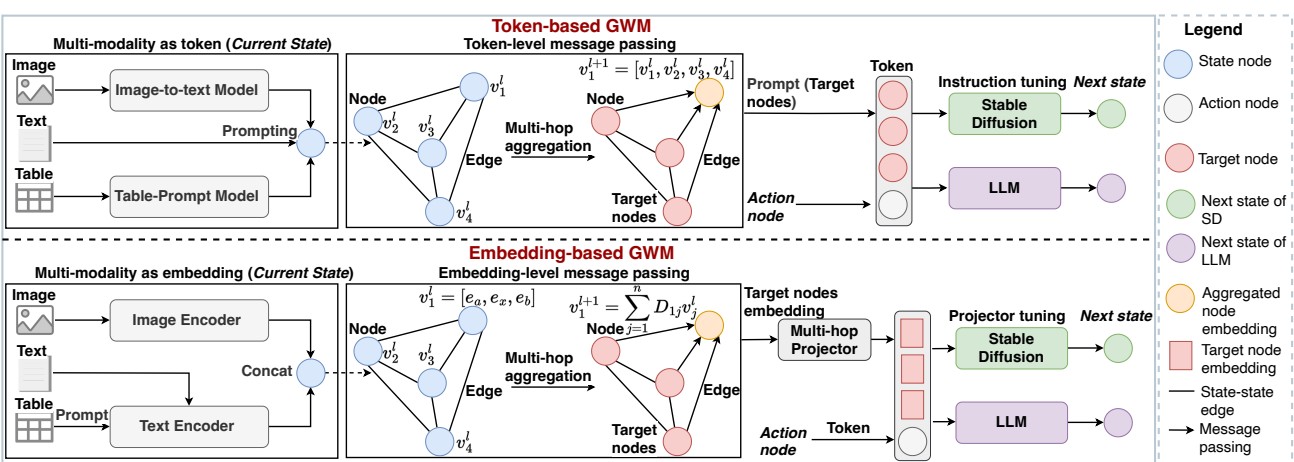

*Figure 3.* **Framework of GWM.** For both token-based and embedding-based GFM, we initially unify the multi-modal current state into graph nodes, conduct message passing, and then combine actions to predict the next state across different modalities through respective decoders. The key distinctions are: 1) Token-based GWM integrates multi-modalities into text, whereas embedding-based GWM uses modality-specific encoders to process them into embeddings; 2) Token-based GWM utilizes text-based methods for message passing, while embedding-based GWM operates at the embedding level; 3) Token-based GWM converts information into prompt form for the decoder, while embedding-based GWM employs a multi-hop projector to manage embedding-level information.

be expressed mathematically as:

$$\mathcal{L} = \mathbb{E}_{\mathbf{z}\sim\text{Enc}(x),c_T,\epsilon\sim\mathcal{N}(0,1),t}\left[\|\epsilon - \epsilon_\theta(\mathbf{z}_t,t,h(c_T))\|^2\right] \quad (2)$$

During each iterative step $t$, a specialized denoising network $\epsilon_\theta(\cdot)$ estimates noise patterns by jointly processing three inputs: the current latent state $\mathbf{z}_t$, a temporal position indicator $t$, and encoded text features $h(c_T)$. The text features $h(c_T) \in \mathbf{R}^{d\times l_{c_T}}$ are extracted using CLIP's text encoder (Radford et al., 2021) $h(c_T) = \text{CLIP}(c_T)$, where $l_{c_T}$ represents the prompt length and $d$ denotes the feature dimensionality.

**Instruction tuning of LLM**. We then design LLM to generate texts for both text and table modalities. We followed the standard instruction tuning practice (Zhang et al., 2023b; Peng et al., 2023), which encourages LLMs to adhere to user requests when returning the outputs. For the input instruction $P_{sa}(\mathbf{h}_{vr}, a)$, we supply a response representing the next predicted states, consisting of $t$ tokens and denoted as $y = \{y_1, \ldots y_t\}$. We train the LLM to yield $f_{\text{SFT}}$ via

$$\mathcal{L}_{SFT} = -\sum_t log P_{f_{\text{SFT}}}(y_t|P_{sa}(\mathbf{h}_{vr},a),y_1,\ldots y_{t-1}). \quad (3)$$

## 4. Embedding-based GFM

Although token-based GFM can relatively simply construct a multi-modal world, it is still limited by the high token cost and a restricted multi-hop field of view. Inspired by stable diffusion, which introduces modeling in latent space rather than directly on pixels to enhance model performance and efficiency, we have introduced embedding-based GFM as shown in the bottom half of Figure 3.

### 4.1. Multi-modality as embedding

The embedding-based GFM unifies the multi-modal nodes in the embedding space. Firstly, as shown in Figure 3, for each modality of the node, we would assign a specific encoder. For the text modality $v^e$, we utilize a BERT model as the encoder $E_b$ to obtain its embedding $e_t$. As for the table node $v^b$, we first transform it into text description as discussed in Section 3.1 and then utilize a BERT model as the encoder $E_b$ to obtain its embedding $e_b$. Finally, we deploy a CLIP $E_c$ model to encode the image node $v^a$ into image embedding $e_a$. We finally obtain the node embedding $e_v = \text{CONCAT}(e_a, e_t, e_b)$ by concatenating the embeddings of all modalities involved with this node. Specifically, if a node misses some modalities, we use zero vectors for them.

### 4.2. Embedding-level message passing

In this section, we first model the relationships between nodes on the graph through multi-hop aggregation, then aggregate the information from different modalities within the nodes through cross-modal fusion into unified embeddings to pass to the subsequent decoders.

**Multi-hop aggregation**. We designed a simplified GCN (Wu et al., 2019; He et al., 2020a) to implement multi-hop aggregation, which directly accomplishes parameter-free feature aggregation at the node feature level. Specifically, for the adjacency matrix $\mathcal{A}$ between nodes, we first normalize it to obtain the matrix $\tilde{\mathcal{A}} = \mathcal{D}^{-\frac{1}{2}}\mathcal{A}\mathcal{D}^{-\frac{1}{2}}$, where $\mathcal{D}$ represents the degree matrix of $\mathcal{A}$. Then, for the node vector $X_e$ composed of all node embeddings $e_v$, we use the obtained normalized adjacency matrix $\tilde{\mathcal{A}}$ to perform $l$-hop

graph aggregation: $X_e^{(l)} = \tilde{\mathcal{A}}^l * X_e$, where $X_e^{(l)}$ is the $l$-hop graph embedding. We retain the embeddings of the first $L$ hops $[X_e, X_e^{(1)}, \ldots, X_e^{(L)}]$ to the subsequent modules.

**Cross-modal fusion**. We further apply a parameterized projector $f_c$ to transform the multi-modalities in the node to unified embeddings: $X_c^{(l)} = f_c(X_e^{(l)})$, where $X_c^{(l)}$ is the unified node embedding. Specifically, we utilize a simple MLP as projector $f_c$ and output the $L$ hops embeddings $X_G = [X_c, X_c^{(1)}, \ldots, X_c^{(L)}]$ to the decoders. Note that GWM-E can be extended to heterogeneous graphs by performing separate multi-hop aggregations for each edge type, followed by flattening the resulting node embeddings into a sequence format suitable for input into the LLM decoder.

### 4.3. Projector tuning

As discussed in Section 3.3, in this section we discuss the tuning of projectors for two different modalities separately.

**Projector tuning of SD**. We first incorporate graph conditioning tokens $h_G(c_G) = X_G$ into the SD models, functioning concurrently with the pre-existing text conditions $h_T(c_T)$: $h(c_T, c_G) = [h_T(c_T), h_G(c_G)] \in \mathbf{R}^{d \times (l_{c_T} + l_{c_G})}$, where $l_{c_G}$ is the length of the graph condition. The training objective then becomes:

$$\mathcal{L} = \mathbb{E}_{\mathbf{z} \sim \text{Enc}(x), c_T, c_G, \epsilon \sim \mathcal{N}(0,1), t} \left[ \| \epsilon - \epsilon_\theta(\mathbf{z}_t, t, h(c_T, c_G)) \|^2 \right]. \tag{4}$$

**Projector tuning of LLM**. We describe the action node $a$ using text as in Section 3.3. We further introduce graph tokens $X_G$ into LLM. The training objective of LLM is to maximize the probability of generating the correct next states. Combining the discussion in Section 3.3, we train the LLM to yield $f_{\text{SFT}}$ via

$$\mathcal{L}_{SFT} = -\sum_t \log P_{f_{\text{SFT}}}(y_t | X_G, a, y_1, \ldots y_{t-1}). \tag{5}$$

We use a training approach similar to prefix tuning (Li & Liang, 2021), where we fix the LLM's parameters and only fine-tune the projector $f_c$'s parameters.

## 5. Experiments

We employ **one unified GWM model across multiple tasks**, comparing its performance against domain-specific methods. Initially, we introduce the tasks within the GWM framework.

**Task description**. The details of the tasks are summarized across three aspects in Table 9 of the Appendix, with further information on tasks and datasets available in Appendix A, and specific action node prompts in Appendix C.

- **World prediction**: It contains three subtasks. **(1) Multi-modal generation and matching (Multi-modal):** We

investigate the node-level multi-modal generation task, where the goal is to predict missing images based on textual captions. We use data from Goodreads (Jin et al., 2024) and the Multi-Modal-Paper dataset (detailed in Appendix A.1). The generated images are evaluated using CLIP Score (Radford et al., 2021) and DINOv2 (Oquab et al., 2023). We compare our approach against several baselines, including Stable Diffusion 1.5 (SD-1.5) (Rombach et al., 2022), its fine-tuned variant (SD-1.5 FT), the image-to-image model ControlNet (Zhang et al., 2023a), and the SOTA INSTRUCTG2I model (Jin et al., 2024). Meanwhile, our edge-level multi-modal matching task evaluates the correspondence between different modalities, using Contrastive MLP (Liu et al., 2022), CLIP (Radford et al., 2021), and fine-tuned CLIP on metrics such as Accuracy, Recall, and F1. Please note that since the multi-modal matching task of Multi-Modal-Paper also includes matching between text and tables, CLIP cannot be applied to this subtask.

**(2) Recommendation (Rec):** As Table 11 of Appendix A.2 illustrates, we utilize three benchmark datasets of varying scales—Baby, Sports, and Clothing—from Amazon's real-world product collections (McAuley et al., 2015). These datasets are commonly used in existing multi-modal graph recommendation systems (Wei et al., 2019a; 2020a). For these edge-level tasks, we benchmark our GWM model against recent state-of-the-art recommendation approaches, including FREEDOM (Zhou & Shen, 2023), as well as representative graph-based models such as LightGCN (He et al., 2020a), MMGCN (Wei et al., 2019b), and GRCN (Wei et al., 2020b). We use Recall and F1 Score as the primary evaluation metrics. **(3) Traditional graph prediction (Graph):** We utilize Cora (Chen et al., 2024b), PubMed (Chen et al., 2024b), and HIV (Wu et al., 2018) datasets. For the Cora and PubMed datasets, we perform node-level and edge-level tasks, while for the HIV dataset, we undertake graph-level tasks. We compare GWM against two traditional graph baselines, GCN (Kipf & Welling, 2016) and GAT (Veličković et al., 2017b), as well as two GFM baselines, LLAGA (Chen et al., 2024a) and OFA (Liu et al., 2023a). We adopt accuracy as the metric. Details can be seen in Appendix A.3.

- **World generation**: It contains two sub-tasks. **(1) Multi-agent collaboration (Multi-agent):** We utilize a multi-modal agent benchmark called AgentClinic (Schmidgall et al., 2024) (in Appendix A.4) to evaluate LLMs within simulated clinical environments. This environment is structured as a graph, with nodes representing different profile-based agents such as patients, measurements, and moderators, and containing various modalities of external knowledge including medical images and patient records. The edges represent interactions between agents and their

*Table 2.* **Multi-modal generation results on Goodreads and Multi-Modal-Paper**. This task is to predict the missing modality based on the given modality. Compared to specific baselines in image generation, GWM achieved the best results.

| Model | Goodreads | | Multi-Modal-Paper | |
|---|---|---|---|---|
| | CLIP | DINOv2 | CLIP | DINOv2 |
| SD-1.5 | 42.16 | 14.84 | 52.62 | 23.64 |
| SD-1.5 FT | 45.81 | 18.97 | 58.49 | 24.13 |
| ControlNet | 42.20 | 19.77 | 52.89 | 24.77 |
| INSTRUCTG2I | **50.37** | **25.54** | 56.37 | 18.80 |
| GWM-T | 47.46 | 20.91 | **59.92** | 23.10 |
| GWM-E | 45.23 | 20.87 | 59.84 | **26.03** |

*Table 3.* **Multi-modal matching results on Goodreads and Multi-Modal-Paper**. It aims to predict the correspondence between different modal-ities. For Goodreads, this task is to predict text-image correspondences. As for Multi-Modal-Paper, it aims to predict text-image, text-table, and table-image correspondences. Note that CLIP and CLIP FT cannot be applied to Multi-Modal-Paper since it includes matching tasks beyond text-image.

| Model | Goodreads | | | Multi-Modal-Paper | | |
|---|---|---|---|---|---|---|
| | Accuracy | Recall | F1 Score | Accuracy | Recall | F1 Score |
| Contrastive MLP | 54.70 | 54.67 | 54.79 | 51.77 | 51.55 | 50.31 |
| CLIP | 83.80 | 83.80 | 83.84 | - | - | - |
| CLIP FT | **92.60** | **92.58** | **92.61** | - | - | - |
| GWM-T | 84.22 | 85.66 | 85.29 | 88.26 | 90.35 | 90.11 |
| GWM-E | 88.82 | 89.73 | 89.06 | **96.23** | **97.21** | **97.13** |

*Table 4.* **Recommendation on Baby, Sports, and Clothing**. Compared with three classical graph baselines, GWM achieved state-of-the-art results on most metrics.

| Model | Baby | | Sports | | Clothing | |
|---|---|---|---|---|---|---|
| | Recall | F1 Score | Recall | F1 Score | Recall | F1 Score |
| FREEDOM | 60.35 | 66.16 | 63.47 | 70.53 | 70.20 | 78.40 |
| LightGCN | 51.11 | 38.22 | 85.36 | **91.32** | 69.08 | 77.21 |
| MMGCN | 57.34 | 61.31 | 61.69 | 68.08 | 64.09 | 71.26 |
| GRCN | 74.35 | 82.47 | 57.31 | 61.23 | 57.60 | 61.74 |
| GWM-T | 70.84 | 75.08 | 84.29 | 88.60 | 71.73 | 74.26 |
| GWM-E | **76.72** | **84.74** | **88.78** | 90.32 | **75.27** | **84.06** |

*Table 5.* **Traditional graph prediction results on Cora, PubMed, and HIV**. It covers representative tasks at the node-level, edge-level, and graph-level. Compared to classic graph baselines and GFM methods, our GWM can match their performance with one unified model.

| Model | Cora | | PubMed | | HIV |
|---|---|---|---|---|---|
| Task Type | Node | Link | Node | Link | Graph |
| GCN | 78.86 | 90.40 | 74.49 | 91.10 | 86.72 |
| GAT | 82.76 | 93.70 | 75.24 | 91.20 | 87.84 |
| LLAGA | **89.22** | 89.18 | **95.03** | 89.18 | 85.42 |
| OFA | 73.21 | 93.12 | 77.80 | **96.39** | 92.04 |
| GWM-T | 81.92 | 88.24 | 92.91 | 91.88 | 92.20 |
| GWM-E | 83.03 | **94.31** | 84.22 | 94.01 | **93.86** |

engagement with knowledge resources. Given the objective of integrating information from all agents to answer medical questions, we define this as a graph-level task. We compare our approach with three LLM-based baselines: CoT (Wei et al., 2022), ToT (Yao et al., 2024), and Few-Shot (Madotto et al., 2021), as well as two additional baselines fine-tuned on the AgentClinic dataset. FT refers to a LLaMA-3-8B model fine-tuned directly on the task. Longformer (Beltagy et al., 2020) is a strong baseline for long-document understanding. We use Accuracy, Recall, and F1 Score as evaluation metrics to assess the correctness of the generated responses. **(2) Retrieval-augmented generation (RAG):** We utilize LongBench v2 (Bai et al., 2024), a benchmark designed for challenging long-context question-answering (in Appendix A.5). Following previous work like GraphRAG (Edge et al., 2024), we divide long context into chunks as nodes of the graph, and the edges between nodes are the similarity of their BERT embeddings. We conduct comparisons with two RAG-based baselines—BM25 (Robertson et al., 2009) and Dragon (Lin et al., 2023)—and three long-context LLMs (128$k$), including Mistral Large 2, Command R+, and GPT-4o mini. Accuracy serves as our evaluation metric.

- **World optimization (Optimization)**: Many existing works (Ho & Ermon, 2016; Hussein et al., 2017; Yang et al., 2024) attempt optimization tasks by imitating the trajectory of expert strategies. Here, we utilize the expert strategy dataset from the text-based embodied task ALF-World (Shridhar et al., 2020; Yang et al., 2024). We model each decision state as graph nodes and derive the edges between nodes based on the similarity of the state images associated with the decisions. We compare GWM with three LLM baselines—COT (Wei et al., 2022), TOT (Yao et al., 2024), and T5 (Raffel et al., 2020) fine-tuned on our dataset (T5 FT) —using BERT-Score (Zhang et al., 2019) (Precision, Recall, and F1 Score) as metrics. Details can be seen in Appendix A.6.

**Implementation details**. We train and test *a single GWM* on all tasks, comparing it with domain-specific baselines for each task. Specifically, for the LLM module, we uniformly use Llama-3-8B, and for stable diffusion, we use SD-v1-5. For the image-to-text model used in GWM-T, we use LLaVA-1.5-7B. The image encoder and text decoder used in GWM-E are CLIP and BERT models, respectively. In addition, our multi-hop projector uses an n-hop MLP to aggregate features from different hops, where n-hop refers to the number of neighborhood hops of the graph nodes used. To ensure the training efficiency of the models, we set the maximum token length for all models at 2k. We use Adam optimizer (Diederik, 2014) for model training and gradually decay the learning rate with LambdaLR scheduler.

All the experiments are conducted on NVIDIA A6000 GPUs. Please refer to Appendix B for other implementation details.

*Table 6.* **Multi-agent collaboration results on AgentClinic**. This task is to answer medical questions by leveraging interactions between agents and external knowledge sources. Compared to classic LLM baselines, GWM-T achieves state-of-the-art results.

| Model | Accuracy | Recall | F1 Score |
|---|---|---|---|
| COT | 45.00 | 33.42 | 32.25 |
| TOT | 35.00 | 33.71 | 29.71 |
| Few-shots | 40.00 | 40.63 | 29.37 |
| Longformer | 25.00 | 20.20 | 14.00 |
| FT | 45.00 | 45.40 | 44.00 |
| GWM-T | **50.00** | **46.42** | **48.20** |
| GWM-E | 45.00 | 39.57 | 35.56 |

## 5.1. A single GWM matches the performance of domain-specific methods across multiple tasks

We train a unified GWM on all tasks and test it across all tasks without further fine-tuning, compared with domain-specific baselines under each task. Specifically, for world prediction, we first report the multi-modal generation and matching results in Table 2 and Table 3. Subsequently, we report the recommendation results in Table 4, and the traditional graph prediction results in Table 5. As for world generation, we report the multi-agent collaboration results in Table 6 and the retrieval-augmented generation results in Table 7. For world optimization, we report the results in Table 8.

We can observe that: (1) *A single GWM* achieves SOTA results in multi-modal generation (Multi-Modal-Paper), multi-agent collaboration, retrieval-augmented generation, as well as planning and optimization, and also performs comparably

*Table 7.* **Retrieval-augmented generation on LongBench v2**. It is a challenging long-context question-answering task that is categorized into easy and hard levels. Compared to classic RAG baselines and LLM models with extended contexts, GWM with limited context length achieved the best results. The result also demonstrates the superiority of GWM-E over GWM-T in tasks involving long contexts.

| Model | Overall | Easy | Hard |
|---|---|---|---|
| BM25 (2k) | 27.45 | **41.18** | 20.59 |
| Dragon (2k) | 23.53 | 35.29 | 17.65 |
| Mistral Large 2 (128k) | 26.31 | 29.42 | 24.45 |
| Command R+ (128k) | 27.43 | 30.19 | 26.32 |
| GPT-4o mini (128k) | 29.01 | 30.23 | 28.03 |
| GWM-T (2k) | 29.40 | 35.71 | 21.74 |
| GWM-E (2k) | **33.32** | 39.16 | **29.52** |

*Table 8.* **Planning and optimization results on ALFWorld**. It is to evaluate how well the methods can imitate the trajectory of expert strategies to effectively assist in solving optimization problems. Compared to classic LLM baselines and text generation baselines, GWM-E has achieved the best results.

| Model | Precision | Recall | F1 Score |
|---|---|---|---|
| Normal | 89.62 | 88.86 | 89.21 |
| COT | 86.87 | 87.74 | 87.27 |
| T5 FT | 92.06 | 91.52 | 91.82 |
| GWM-T | 88.10 | 87.05 | 87.42 |
| GWM-E | **93.27** | **92.36** | **92.13** |

to domain-specific baselines in other tasks. This demonstrates GWM's ability to generalize and its applicability across a broad range of tasks. (2) GWM demonstrates promising capabilities in some highly challenging tasks, such as long-context RAG (shown in Table 7). GWM with a context length of 2k can outperform LLM models with a context length of 128k in RAG tasks, showcasing GWM's potential in understanding and reasoning with long texts. (3) The design of the latent embedding enables GWM-E to outperform GWM-T in five out of seven tasks with approximately 5-10 times fewer token costs. This demonstrates the efficiency and effectiveness of embedding-based message passing.

## 5.2. GWM benefits from multi-hop graphs

To explore whether multi-hop graphs can enhance the performance of GWM, we compared the effectiveness of four different hop settings with a no-graph baseline using GWM-E on six tasks, as illustrated in Figure 4. Specifically, we measured the average performance across five settings for all tasks. For Multi-modal tasks, we use DINOv2 and F1 Score to calculate average performance, while for Rec, Agent, and Optimization tasks, we exclusively use the F1 Score. Accuracy metrics were employed for the remaining tasks. Graphs have consistently enhanced GWM-E performance across all tasks, showing a minimum relative gain of 20% on graph-related tasks. However, an increased hop number does not always lead to better performance since it can cause over-smoothing and introduce redundant information.

## 5.3. GWM boosts zero-shot/few-shot performance

To validate the zero-shot/few-shot capabilities of GWM, we conduct experiments with GWM-E and GWM-T on the Agent and RAG tasks, as shown in Figure 5 ("-T" and "-E" respectively represent the experimental results of GWM-T and GWM-E). Here, Single Data refers to training GWM solely on the Agent or RAG task. Zero-shot refers to training GWM on tasks other than Agent or RAG and testing it on Agent or RAG. Fine-tuned GWM refers to training

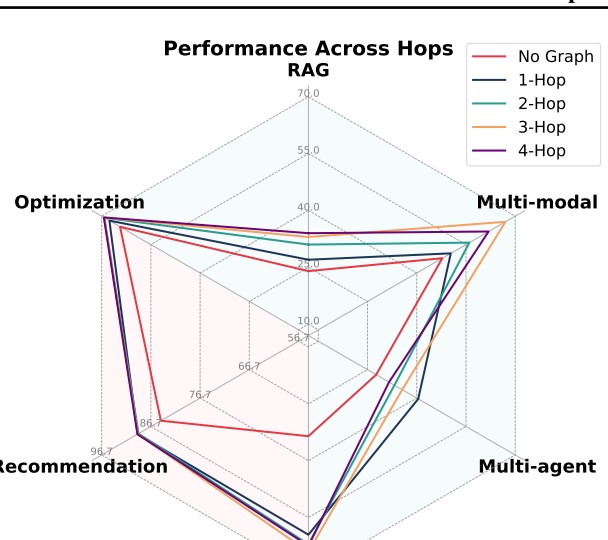

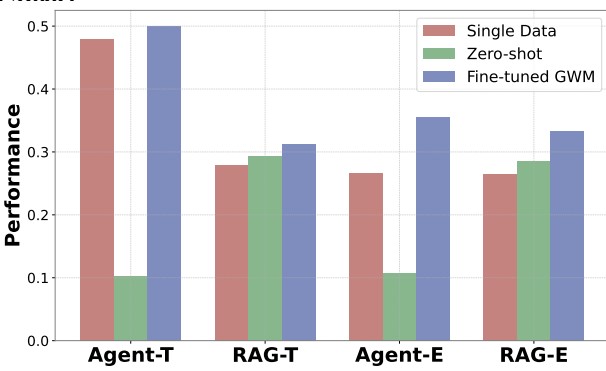

*Figure 4.* **Multi-hop graphs enhance GWM's performance across representative tasks in six domains**. We can observe that the introduction of graphs has benefited GWM-E across all tasks compared to no graph. Moreover, excessive hops can lead to over-smoothing, thereby decreasing performance.

*Figure 5.* **GWM boosts zero-shot/few-shot performance on multi-agent collaboration (Agent) and retrieval-augmented generation (RAG) tasks**. Note that "-T" and "-E" respectively represent the experimental results of GWM-T and GWM-E. It can be observed that GWM can quickly adapt to new tasks with a small amount of domain-specific training data. Moreover, GWM's strong generalization ability can boost the performance of Agent and RAG.

GWM on tasks other than Agent or RAG, followed by few-shot fine-tuning with 10% of the data from Agent or RAG. We can observe that GWM adapts effectively to new tasks using only a small amount of domain-specific training data. Moreover, we observe that the zero-shot results of GWM on the RAG task are even better than those from Single Data, indicating that GWM's strong generalization ability greatly benefits tasks with limited training data like Agent or RAG.

## 6. Additional Related Work

**Graph for Modelling Relations**. Graphs are highly effective in modeling complex relationships (Fey et al., 2023; Cao et al., 2023; Gao & Xu, 2020; Chen et al., 2022; Wu et al., 2022; Yang et al., 2021), extracting nodes and edges to model relational data with embeddings. Graph Neural Networks (GNNs) (Kipf & Welling, 2017; Hamilton et al., 2017a; Veličković et al., 2017a; Schlichtkrull et al., 2017) have emerged as a dominant approach, particularly in recommendation systems (Min et al., 2022) and social networks (Wu et al., 2020). To further address the vast array of tasks and data, scholars have proposed the GFM (Chen et al., 2024a; Liu et al., 2023a) to explore GNNs' zero-shot or few-shot capabilities (Fey et al., 2023; Cao et al., 2023; Gao & Xu, 2020; Chen et al., 2022) to tackle challenges such as the cold start problem in recommendations.

**World Model**. The WM (Ha & Schmidhuber, 2018) is to construct the world observations as states and predict future states based on given actions. Existing WMs (Wu et al., 2024a; Bruce et al., 2024) primarily focus on how to utilize unstructured data to predict state transitions, thereby enhancing the effectiveness of sequence generation tasks. Genie (Bruce et al., 2024) trained a foundation world model using a massive amount of unlabelled, serialized internet videos, which has provided benefits for the planning outcomes of downstream tasks. Additionally, some WMs (Zhang et al., 2021; Zhu et al., 2022) have attempted to integrate structured data with GWM. $L^3P$ (Zhang et al., 2021) uses graphs to model each step of the agent's decision-making process and their connections, thus enhancing scalable planning in reinforcement learning. However, they are still largely confined to planning and optimization scenarios, which limits their potential for task generalization as WMs. Thus we develop GWM that integrates the capabilities of graphs with WM to generalize across diverse tasks.

## 7. Conclusion

We propose GWM, a unified framework that uses a graph world state to tackle diverse prediction, generation, and planning tasks. Across six benchmarks, GWM matches domain-specific baselines while benefiting from multi-hop graph structures, showing strong generality and flexibility. It also improves zero-shot and few-shot performance, indicating strong cross-task generalization. GWM currently supports text, table, and image modalities, with plans to extend to more. While the current implementation focuses on homophilous graphs, we aim to expand it to support both homophilous and non-homophilous structures for broader applicability. Its modular design also makes it a flexible base for future multi-modal graph reasoning tasks.

## Impact Statement

The WM serves as a unified framework for prediction, generation, and decision-making across various applications. While traditional WMs are constrained to single-modality and unstructured data, our proposed GWM enhances them by embedding-level message passing and aggregation to integrate structured and multi-modal data, bridging the gap between unstructured and structured processing. GWM demonstrates significant potential as a foundational graph-based model for real-world multi-modal tasks; however, its current scope is limited by the number of supported modalities and the simplicity of its graph architecture, necessitating further advancements for broader applicability and enhanced relational modeling. Future applications should also prioritize ethical considerations, recognizing that efforts are needed to ensure that GWM's responses are reliable, unbiased, and safe in real-world deployments, thereby preventing potential harm to users. In addition, the data utilized in this work are collected in compliance with applicable laws and licensing agreements. Their usage is also transparent and harmless.

The security of Large Language Models (LLMs) has always been a concern. Unfortunately, current LLMs sometimes produce harmful and biased information unexpectedly. Our proposed method uses LLMs to generate simulated queries and summary responses, which are only used to construct a graph of records and connect text chunks from long documents. However, more work is needed in real-world applications to ensure that LLMs' responses are reliable and harmless, so that they do not harm users.

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

# A. More on GWM Task

This section discusses the detailed processing procedure for each dataset collected in GWM. We summarize their general information in Table 9.

## A.1. Multi-modal generation and matching

**Dataset descriptions**. In this study, we utilize two datasets for multi-modal generation and matching: the Goodreads dataset (Wan et al., 2019) and our curated Multi-Modal-Paper dataset.

**(1) Goodreads**: The Goodreads dataset is a large-scale collection of book-related metadata, textual descriptions, and cover images, widely used in prior multi-modal research (Jin et al., 2024). The Goodreads dataset is structured as a graph, where each book is represented as a node, and edges signify similar-book semantics.

**(2) Multi-Model-Paper**: Multi-Modal-Paper dataset is a curated dataset of academic papers, incorporating textual content, figures, tables, and metadata to facilitate research on scholarly document analysis. The raw LaTeX files of the papers were collected from ArXiv[1] using the ArXiv API, in accordance with the papers' licenses, which are primarily CC0 or CC-BY 4.0, permitting redistribution and sharing. Using carefully selected survey papers from various domains of Artificial Intelligence (AI), including Natural Language Processing (NLP), Computer Vision (CV), Bioinformatics, and Robotics, as seed papers, we employ a breadth-first search (BFS) algorithm to gather cited papers. Then, we traverse through the abstract syntax tree built from the LaTeX file to extract graph-structured multi-modal data for each gathered paper.

For the multi-modal generation task, we sample figure-caption pairs for training and evaluation. GWM and baseline models are tasked with reconstructing the figures from the provided captions. For the multi-modal matching task, we sample cited-citing pairs using reference relationship, namely the macro command "\ref". The cited-citing pairs are usually figure-text or table-text pairs.

A detailed statistical overview of the Goodreads and Multi-Modal-Paper datasets is presented in Table 10.

**Baselines details**. For multi-modal generation tasks, we employ two text-to-image baseline models, SD-1.5 and SD-1.5 FT, along with an image-to-image baseline model, ControlNet.

- **SD-1.5**: A pre-trained Stable Diffusion v1.5 model (Rombach et al., 2022) used for text-to-image generation without task-specific fine-tuning.

---

[1]https://arxiv.org/ We thank ArXiv for providing open access interoperability.

*Table 9.* **Detailed summarization of all collected datasets in GWM.** We summarize the dataset names, tasks, action level, multi-modality, nodes, and edges in the table.

| Dataset | Task | Action Level | Multi-modality | Nodes | Edges |
|---|---|---|---|---|---|
| Goodreads | Multi-modal generation/matching | Node level | Text/image | Text/image nodes | Similar-book semantic |
| Multi-Modal-Paper | Multi-modal generation/matching | Node level | Text/image/table | Text/image/table nodes | References |
| Baby | Recommendation | Link level | Text/table/image | User/item nodes | User-item interactions |
| Sports | Recommendation | Link level | Text/table/image | User/item nodes | User-item interactions |
| Clothing | Recommendation | Link level | Text/table/image | User/item nodes | User-item interactions |
| Cora | Traditional graph prediction | Node/edge level | Text | Research paper nodes | Citation |
| PubMed | Traditional graph prediction | Node/edge level | Text | Research paper nodes | Citation |
| HIV | Traditional graph prediction | Graph level | Text | Atoms nodes | Atoms bonds |
| AgentClinic | Multi-agent collaboration | Graph level | Text/image | Agent/image/text nodes | Agent-agent/image/text |
| LongBench v2 | Retrieval-augmented generation | Unintended action | Text | Chunk nodes | Chunk-chunk similarity |
| ALFWorld | Planning and optimization | Graph level | Text/image | State nodes | State images similarity |

*Table 10.* **Data statistics for multi-modal generation and matching.** In Multi-Modal-Paper, there are 58565 text-nodes, 7380 figure-nodes, and 6792 table-nodes.

| Dataset | #Node | #Edges |
|---|---|---|
| **Goodreads** | 93,475 | 637,210 |
| **Multi-Modal-Paper** | 72,737 | 51,840 |

*Table 11.* **Data statistics for recommendation.** It includes three datasets of different scales, with the sizes ranging from small to large as follows: Baby, Sports, and Clothing.

| Dataset | #User | #Item | #Edges | Sparsity |
|---|---|---|---|---|
| **Baby** | 19,445 | 7,050 | 160,792 | 99.883% |
| **Sports** | 35,598 | 18,357 | 296,337 | 99.955% |
| **Clothing** | 39,387 | 23,033 | 278,677 | 99.969% |

- **SD-1.5 FT**: Stable Diffusion v1.5 models, each fine-tuned separately on the training splits of the Goodreads and Multi-Modal-Paper datasets.

- **ControlNet**: An extension of Stable Diffusion that incorporates structural guidance, such as edge maps or depth maps, to enhance control over generated images (Zhang et al., 2023a).

We use three baselines for multi-modal matching: Contrastive MLP, CLIP, and CLIP FT. Since the vertices of the edge in the Multi-Modal-Paper dataset can include tables, namely the text-table pair that are not suitable for the CLIP model, we exclusively use Contrastive MLP for this dataset.

- **Contrastive MLP** (Liu et al., 2022): A multi-layer perceptron trained to predict multi-modal matching by processing embeddings from different modalities. The text and table embeddings are encoded by BERT while the image embedding is encoded by CLIP. Embeddings from different modalities are padding to the same dimension.

- **CLIP**: A pre-trained vision-language model designed for image-text alignment, using contrastive learning to map corresponding image and text embeddings into a shared space (Radford et al., 2021).

- **CLIP FT**: A fine-tuned version of CLIP, adapted to the specific dataset to enhance multi-modal matching performance.

## A.2. Recommendation

**Dataset descriptions**. In the recommendation task, we conduct extensive evaluations using three Amazon datasets extensively recognized in prior research (McAuley et al., 2015), specifically: Baby, Sports, and Outdoors, as well as Clothing Shoes, and Jewelry. For simplicity, these datasets are hereafter referred to as Baby, Sports, and Clothing, respectively. Utilizing the 5-core setting, we filter inactive users and items with less than five interactions. Each dataset encompasses both visual and textual modalities and we use the extracted visual and textual features from existing work (Zhou, 2023). Here the visual modality is the product image and the textual modality is the product description. The characteristics of these datasets are shown in Table 11.

**Baselines details**. We compare GWM with three representative GNN baselines.

- **LightGCN**: Employs a simplified graph convolutional network to learn user and item interaction graph (He et al., 2020b).

- **MMGCN**: Learns user preferences across multiple modalities via message-passing on modality-specific user-item graphs, improving recommendations in multimedia contexts (Wei et al., 2019a).

- **GRCN**: Refines interaction graphs using multimedia content to identify and remove noisy edges, thereby sharpening the recommendation process (Wei et al., 2020a).

## A.3. Traditional graph prediction

**Dataset descriptions**. In the traditional graph prediction task, we evaluate GWM on Cora, PubMed, and HIV datasets. (1) Cora (Chen et al., 2024b): Cora is a citation network in the computer science domain, where nodes represent research papers and edges denote citation relationships. Each node includes the paper's title and abstract as text features, with labels indicating paper categories. Tasks on Cora include category prediction (node level) and citation link identification (link level). (2) PubMed (Chen et al., 2024b): PubMed is a biomedical citation network, similar to Cora, with nodes representing papers and edges indicating citation relationships. (3) HIV (Liu et al., 2023a): HIV is a molecular dataset constructed from MOLHIV dataset (Wu et al., 2018) that contains over 40,000 compounds annotated for their ability to inhibit HIV replication. Molecular structures and graph representations are generated from SMILES strings, with atoms (nodes) and bonds (edges) described using natural language.

**Baselines details**. The settings for the baselines primarily follow LLAGA (Chen et al., 2024a) and OFA (Liu et al., 2023a). We convert all nodes and labels in the Cora, PubMed, and HIV datasets into text. For all methods, we use BERT to obtain text embedding. We divide all datasets into training, validation, and test sets in an 8:1:1 ratio.

- **GCN** (Kipf & Welling, 2017): Applies spectral-based convolution operations to capture local graph structures and propagate information across nodes, serving as a fundamental baseline for graph-based learning.

- **GAT** (Veličković et al., 2017a): Enhances node representation learning by incorporating attention mechanisms, allowing adaptive weighting of neighboring nodes to improve feature aggregation.

- **LLAGA** (Chen et al., 2024a): Integrates LLM with graph structures to enhance reasoning and information retrieval in multi-modal and structured data scenarios.

- **OFA** (Liu et al., 2023a): Unifies vision, language, and multi-modal learning tasks within a single framework, leveraging pre-trained knowledge to facilitate cross-modal understanding and adaptation.

## A.4. Multi-agent collaboration

**Dataset descriptions**. In the multi-agent collaboration task, we evaluate GWM on AgentClinic (Schmidgall et al., 2024) benchmark, specifically AgentClinic-NEJM collected from the New England Journal of Medicine (NEJM) case challenges. Each case in AgentClinic-NEJM is multimodal, comprising a case description, patient profile, clinical photograph, measurement results, and five candidate diagnoses. We partition AgentClinic-NEJM into training, validation,

and test sets using a 4:1:1 split ratio. To simulate real-world clinical procedures, we employ the simulated clinical environments from AgentClinic to gather dialogues between the patient and doctor, along with physical examination results. This environment is modeled as a graph, where nodes represent various profile-based agents and edges capture the interactions between agents and their engagement with knowledge resources. We utilize Meta-Llama-3-70B-Instruct[2] as backbone model for simulation. In the final diagnosis procedure, we apply both GWM and LLM baselines for comparison, where the graph information is converted into textual format before being processed by LLM baselines.

**Baselines details**. We compare GWM with three classic LLM baselines adopting different reasoning strategies. We use Meta-Llama-3-8B-Instruct[3] as the backbone model to align with GWM.

- **CoT**: Adopts Chain-of-Thought (Wei et al., 2022) prompting, which enhances reasoning by decomposing complex problems into intermediate steps, improving performance on multi-step reasoning tasks.

- **ToT**: Adopts Tree-of-Thought (Yao et al., 2024) prompting, which explores multiple reasoning paths in a tree-like structure, enabling iterative evaluation and refinement for more robust decision-making.

- **Few-shots**: Adopts Few-shot (Brown et al., 2020) prompting, where the model is provided with a limited number of in-context examples to guide task-specific reasoning without requiring fine-tuning.

## A.5. Retrieval-augmented generation

The purpose of Retrieval-Augmented Generation (RAG) is to enhance the generation capabilities of Large Language Models (LLMs) by retrieving information from external knowledge (Lewis et al., 2020; Gao et al., 2023; Zhao et al., 2024). We introduce its dataset and baselines as follows:

**Dataset descriptions**. We employ LongBench v2 (Bai et al., 2024), a benchmark specifically designed to test long-context understanding and reasoning. This benchmark comprises 503 challenging multiple-choice questions, with contextual lengths ranging from 8,000 to 2 million words, spanning six major task categories: Single-Doc QA, Multi-Doc QA, Long In-context Learning, Long-dialogue History Understanding, Code Repository Understanding, and Long Structured Data Understanding. The questions are stratified into easy and hard levels based on the difficulty encountered by human experts and models during their resolution. In

---

[2]https://huggingface.co/meta-llama/Meta-Llama-3-70B-Instruct

[3]https://huggingface.co/meta-llama/Meta-Llama-3-8B-Instruct

alignment with methodologies from previous studies such as GraphRAG (Edge et al., 2024), we segment long contexts into chunks that serve as graph nodes, with edges defined by the similarity of their BERT embeddings. Building on this, we select the Top-k (k=5 in our setting) chunks with the highest similarity to the question's embedding to feed into the GWM. For this task, we divided the dataset into training, validation, and test sets in an 8:1:1 ratio.

**Baselines details**. We conduct comparisons with two RAG-based baselines—BM25 (Robertson et al., 2009) and Dragon (Lin et al., 2023)—and three long-context LLMs ($128k$), including Mistral Large 2[4], Command R+[5], and GPT-4o mini[6]. Their details are as follows:

- **BM25**: A widely-used ranking function in information sparse retrieval. It inputs the retrieved context along with the question into the Llama-3-8B model to generate a response.

- **Dragon**: It employs contrastive learning and other training tricks to finetune its ability to retrieve memory chunks. Using the Llama-3-8B model, it processes the retrieved context and the question to produce a response.

- **Mistral Large 2**: Mistral Large 2 from Mistral AI boasts 123 billion parameters, with a context limit of 128 k tokens. This model is one of the largest currently available, offering exceptional depth in language understanding and generation capabilities, suited for tackling the most demanding NLP tasks across various domains.

- **Command R+**: Command R+ by Cohere is a massive language model with 104 billion parameters, also supporting a context size of up to 128 k tokens. It is optimized for understanding and executing complex commands, making it particularly effective in interactive applications where precise and nuanced language comprehension is critical.

- **GPT-4o mini**: GPT-4o mini, developed by OpenAI, is a variant of the GPT-4 series. Unlike its larger counterparts, specific details about the model's size in terms of parameters are not provided, but it is designed to handle a maximum context size of 128k tokens. This model is geared towards applications requiring high-quality text generation with potentially limited computational resources.

### A.6. Planning and optimization

This task is designed to measure how effectively different methods can imitate the trajectory of expert strategies, which is very helpful for planning and optimization tasks.

---

[4]https://huggingface.co/mistralai/Mistral-Large-Instruct-2407
[5]https://huggingface.co/CohereForAI/c4ai-command-r-plus
[6]https://openai.com/index/gpt-4o-mini-advancing-cost-efficient-intelligence/

**Dataset descriptions**. We employ the expert strategy dataset from the text-based embodied task framework, ALFWorld (Shridhar et al., 2020; Yang et al., 2024). This dataset provides detailed descriptions of the expert's strategic state at each decision point, incorporating both images and text, along with the corresponding decisions made in text format. In our approach, we represent each decision state as nodes within a graph and establish edges between these nodes based on the similarity of the state images linked to each decision. Our total sample size is 10,000, and it is divided into training, validation, and test sets in an 8:1:1 ratio.

**Baselines details**. For all baselines, we first use LLaVA-1.5-7B to convert the image of each state into a text description.

- **Normal**: It directly inputs the text description of the current state into Llama-3-8B to get the response.

- **COT**: It adopts Chain-of-Thought (Wei et al., 2022) prompting into baseline Normal to enhance reasoning ability when predicting.

- **T5 FT (Raffel et al., 2020)**: It is a versatile language model designed by Google Research, which treats every language problem as a text-to-text task, enhancing its adaptability across a broad range of NLP applications. Here we finetune it on the dataset of this task.

## B. Hyper-parameters

For the GWM-E, we employ an n-hop MLP, where for the LLM decoder each MLP has dimensions of 2048*4096, and for the SD decoder, each MLP has dimensions of 2048*768. We fix the parameters of LLM and only fine-tune the parameters of MLP. For GWM-T, we select a maximum of 2k token-limited hops for each task to ensure a balance between efficiency and performance. We apply Lora (Lora rank = 8) (Hu et al., 2021) for efficient training. We have summarized the hyperparameters for training different models in Table 12.

*Table 12.* Hyper-parameter configuration for model training.

| Parameter | GWM-T LLM | GWM-T SD | GWM-E LLM | GWM-E SD |
|---|---|---|---|---|
| Optimizer | AdamW | AdamW | AdamW | AdamW |
| Adam $\epsilon$ | 1e-8 | 1e-8 | 1e-8 | 1e-8 |
| Adam $(\beta_1, \beta_2)$ | (0.9, 0.999) | (0.9, 0.999) | (0.9, 0.999) | (0.9, 0.999) |
| Weight decay | 1e-2 | 1e-2 | 1e-2 | 1e-2 |
| Batch size per GPU | 4 | 1 | 10 | 16 |
| Gradient Accumulation | 8 | 4 | 1 | 4 |
| Epochs | 4 | 5 | 1 | 30 |
| Resolution | - | 512 | - | 256 |
| Learning rate | 3e-4 | 1e-5 | 3e-4 | 1e-5 |
| Backbone SD | Llama-3-8B | SD-v1-5 | Llama-3-8B | SD-v1-5 |

## C. Prompt Usage of GWM

We summarize all the prompts we used in GWM in this section. We first introduce the action prompts in GWM.

Specifically, we have summarized the action prompts for multi-modal generation and matching in Tables 13 and 14. The action prompts for recommendations are summarized in Table 15. Additionally, the action prompts for traditional graph prediction are outlined in Tables 16, 17, 18, and 19. Moreover, we have also summarized the action prompts for multi-agent collaboration, retrieval-augmented generation, and planning and optimization in Tables 20, 21, and 22. Then, we introduce prompts used in GWM-T. We summarize the prompt $P_u$ of multi-modality as tokens in GWM-T in Table 23. Moreover, we introduce the prompt $f_v(\cdot)$ of aggregating central node and neighbor nodes in GWM-T in Table 24.

## D. Qualitative Comparisons for All Tasks of GWM

These tables present comprehensive qualitative comparisons across all task categories evaluated in our GWM framework study. Each table demonstrates the superior performance of our Graph World Model (GWM) variants compared to state-of-the-art baselines through concrete examples. Table 25 showcases multi-modal generation capabilities where GWM-T successfully predicts missing modalities from given inputs. Table 26 illustrates multi-modal matching tasks where GWM-E accurately determines correspondence between different modalities. Table 27 demonstrates recommendation performance where GWM-E correctly predicts user-item connections. Table 28 highlights traditional graph prediction tasks where GWM-E excels in node classification. Table 29 presents multi-agent collaboration scenarios where GWM-T integrates multiple agent contexts for medical diagnosis. Table 30 shows retrieval-augmented generation capabilities where GWM-E effectively combines retrieved documents with user queries. Finally, Table 31 demonstrates planning and optimization tasks where GWM-E predicts optimal decision-making behaviors in embodied environments. These qualitative results consistently validate the effectiveness of our approach across diverse task domains.

## E. Training and Inference Efficiency

Accurately comparing the training and inference efficiency of GWM with other FMs is highly challenging because many FMs are not designed to address multimodal problems and utilize various architectures. We can only compare the efficiency between GWM and LLM-based FMs from principles. For GWM-T, its efficiency shows no fundamental difference from other LLM-based FMs, as both are based on the standard instruction tuning. For GWM-E, its training process only requires fine-tuning the projector as stated in section 4.3, and its embedding-based method also saves a significant amount of token cost, making it more efficient.

For GWM-E, it takes approximately 7 hours ($\sim \frac{1}{4}$ of GWM-T) of training time on four NVIDIA A6000 GPUs described in implementation details of section 5, and the inference time per case averages 0.213s (similar to GWM-T). Moreover, GWM-E significantly reduces memory usage with a shorter token length of 140.23 ($\sim \frac{1}{14}$ of GWM-T).

*Table 13.* **Action prompt of multi-modal generation**.

This is a multi-modal generation task. Please predict the missing modality based on the given modality: {modality}.

*Table 14.* **Action prompt of multi-modal matching**.

This task involves matching multi-modal information. Given two modalities: {modality 1} and {modality 2}, please determine whether they correspond with each other.

*Table 15.* **Action prompt of recommendation**.

This is a recommendation task. Given the user node and item node: {user node} and {item node}, please tell me whether these two nodes should connect to each other.

*Table 16.* **Action prompt of node classification of Cora**.

Given a node-centered graph: {node}, each node represents a paper, we need to classify the center node into 7 classes: Case Based, Genetic Algorithms, Neural Networks, Probabilistic Methods, Reinforcement Learning, Rule Learning, Theory, please tell me which class the center node belongs to?

*Table 17.* **Action prompt of node classification of PubMed**.

Given a node-centered graph: {node}, each node represents a paper about Diabetes, we need to classify the center node into 3 classes: Diabetes Mellitus Experimental, Diabetes Mellitus Type 1, and Diabetes Mellitus Type 2, please tell me which class the center node belongs to?

*Table 18.* **Action prompt of link prediction of Cora and PubMed**.

Given two nodes information: {node 1} and {node 2}, please tell me whether two center nodes in the subgraphs should connect to each other.

*Table 19.* **Action prompt of graph classification of HIV**.

Human immunodeficiency viruses (HIV) are a type of retrovirus, which induces acquired immune deficiency syndrome (AIDs). Please determine whether this molecule {molecule} is effective for this assay.

*Table 20.* **Action prompt of multi-agent collaboration**.

| |
|---|
| This is a Multi-Agent Collaborative Generation task for creating dynamic conversational interactions. Given a user query: {user query} and context of three distinct agents: {Patient Agent Context}, {Measurement Agent Context}, and {Moderato Agent Context}, Please generate a well-rounded response to the user's question. |

*Table 21.* **Action prompt of retrieval-augmented generation**.

| |
|---|
| This is a Retrieval-Augmented Generation task for improving response quality in dialogue systems. Given a user query: {user query} and a set of retrieved documents: {retrieved documents}, the goal is to generate a coherent and contextually relevant response. Please generate a response that integrates information from the retrieved documents to accurately address the user's query. |

*Table 22.* **Action prompt of planning and optimization**.

| |
|---|
| This is an embodied household task, please predict the next decision-making behavior based on multimodal historical information: {historical information}. |

*Table 23.* **Prompt $P_u$ of multi-modality as token in GWM-T**.

| |
|---|
| The image's text description is: {image's text description}, original text is: {original text}, table description is: {table description}. |

*Table 24.* **Prompt $f_v(\cdot)$ of aggregating central node and neighbor nodes in GWM-T**.

| |
|---|
| The text description of the central node is: {center node}, and the text descriptions of the neighboring nodes are: {neighbor nodes}. |

*Table 25.* **Task description and output comparison of multi-modal generation.** This task is to predict the missing modality based on the given modality. Here we utilize one case of Goodreads dataset as examples. We show the output results of GWM-T, the best performing GWM, and ControlNet , the strongest baseline.

| Task Description | | |
|---|---|---|
| **Task Name** | Multi-modal generation | |
| **Given modality** | **Action prompt** | **Ground truth** |
| Title: The Shark-Infested Custard | This is a multi-modal generation task. Please predict the missing modality based on the given modality: {modality}. |  |
| **Method** | **Output Results** | |
| ControlNet |  | |
| GWM-T |  | |

*Table 26.* **Task description and output comparison of multi-modal matching.** This task is to predict whether two modalities correspond with each other. Here we utilize one case of Multi-Modal-Paper dataset as examples. We show the output results of GWM-E, the best performing GWM, and Contrastive MLP, the strongest baseline. Note that modality can be an image, a table, or a text.

| Task Description | | | |
|---|---|---|---|
| **Task Name** | | Multi-modal matching | |
| **Modality 1: figure** | **Modality 2: text** | **Action prompt** | **Ground truth** |
|  | Illustration of different formats of STL expressions. (a) Different expression formats of the same STL. (b) The binary tree representation of STL. | This task involves matching multi-modal information. Given two modalities: {modality 1} and {modality 2}, please determine whether they correspond with each other. | yes |
| **Method** | | **Output Results** | |
| Contrastive MLP | | no | |
| GWM-E | | yes | |

*Table 27.* **Task description and output comparison of recommendation.** This task is to predict whether the user node and the item node are connected. Here we utilize one case of Baby dataset as examples. We show the output results of GWM-E, the best performing GWM, and LightGCN, the strongest baseline.

| Task Description | | | |
|---|---|---|---|
| **Task Name** | | Recommendation | |
| **User node** | **Item node** | **Action prompt** | **Ground truth** |
| User online product reviews series: I struggled to find full slips, especially larger ones. The first one was too small; the second fit well and was affordable. The beads looked stunning, perfect for beadwork. The earrings broke immediately due to poor quality. I love the 3 flower sister Hawaii glass beads on my Pandora bracelet. They're pretty and large, and my eight-year-old finds them comfy enough to sleep in ... |  | This is a recommendation task. Given the user node and item node: {user node} and {item node}, please tell me whether these two nodes should connect to each other. | no |
| **Method** | | **Output Results** | |
| LightGCN | | yes | |
| GWM-E | | no | |

*Table 28.* **Task description and output comparison of traditional graph prediction.** This task aims to perform predictions at three different levels: node, link, and graph. Here we utilize one case of Cora's node prediction. We show the output results of GWM-E, the best performing GWM, and LLAGA, the strongest baseline.

| Task Description | | |
|---|---|---|
| **Task Name** | Traditional graph prediction | |
| **Node** | **Action prompt** | **Ground truth** |
| Learning under persistent drift: In this paper we study learning algorithms for environments which are changing over time. Unlike most previous work, we are interested in the case where the changes might be rapid but their "direction" is relatively constant. We model this type of change by assuming that the target distribution is changing continuously at a constant rate from one extreme distribution to another. We show in this case how to use a simple weighting scheme to estimate the error of an hypothesis, and using this estimate, to minimize the error of the prediction. | Given a node-centered graph: {node}, each node represents a paper, we need to classify the center node into 7 classes: Case Based, Genetic Algorithms, Neural Networks, Probabilistic Methods, Reinforcement Learning, Rule Learning, Theory, please tell me which class the center node belongs to? | Theory |
| **Method** | **Output Results** | |
| LLAGA | Neural Networks | |
| GWM-E | Theory | |

Table 29. **Multi-agent collaboration task and output comparison.** Disease-related query answering through agent interaction and external knowledge. Results show GWM-T vs COT baseline. **Note: Medical images may cause discomfort but are from real datasets.**

| Task Description | | |
|---|---|---|
| **Task** | Multi-agent collaboration | |
| **User Query** | **Patient Agent** | **Moderator Agent** |
| 53-year-old man, 3-year history: itchy rash, Raynaud's, dysphagia, burning hands. Exam: firm papules on forehead with glabellar grooves, waxy papules on hands with thickening and contractures. Similar changes on nose, lips, ears, trunk, feet. No telangiectasia/calcinosis. Sensory neuropathy in hands/arms/face. Normal thyroid. IgG-monoclonal gammopathy, normal bone marrow. Choices: (A) AL amyloidosis (B) Multiple myeloma (C) Scleredema (D) Scleromyxedema (E) Systemic sclerosis | Role: 53-year-old patient with 3-year symptoms including itchy rash, firm forehead papules causing brow grooves, waxy hand papules with thickening and finger contractures. Experience Raynaud's phenomenon, dysphagia, burning hands, and numbness in hands/arms/face. Aware of normal thyroid tests and abnormal blood protein but unaware of diagnosis implications. | Moderator organizing case information: Test results: Normal thyroid function, IgG-monoclonal gammopathy detected, normal bone marrow biopsy. Key findings: Extracellular yellow-brown deposits in dermis on skin biopsy. |
| **Measurement Agent** | **Action Prompt** | **Ground Truth** |
|  | Multi-Agent Collaborative Generation task for dynamic conversational interactions. Generate response using Patient, Measurement, and Moderator agent contexts for the given user query. | D |
| **Method** | **Output Results** | |
| COT | Analysis of patient symptoms and test results suggests monoclonal gammopathy with characteristic skin findings. DIAGNOSIS: Multiple myeloma (B) | |
| GWM-T | D | |

*Table 30.* **Task description and output comparison of retrieval-augmented generation.** This task is to generate a response that integrates information from the retrieved documents to accurately address the user's query. Here we utilize one case of LongBench v2 dataset as examples. We show the output results of GWM-E, the best performing GWM, and GPT-4o mini, the strongest baseline.

| Task Description | | | |
|---|---|---|---|
| **Task Name** | | Retrieval-augmented generation | |
| **User query** | **Document** | **Action prompt** | **Ground truth** |
| What is the correct answer to this question: You are given a grammar book of Kalamang language, now translate the following Kalamang sentence into English: Faisal emun me mindi don bolonet me ma he kademor. Choices: (A) Faisal's mother is still angry at him for a little thing like that. (B) Faisal's mother turns furious at him for a big thing like that. (C) Faisal's mother gets frustrated at him for a big thing like this. (D) Faisal's mother gets angry at him for a little thing like that. Format your response as follows: "The correct answer is (insert answer here)". | There are very few households with two fluent Kalamang-speaking parents and children born after 1990, but even in those households the children are not raised in Kalamang. As indicated above, non-fluent speakers have a good passive command of Kalamang ... Fluent Kalamang speakers do not necessarily shift to Papuan Malay when they join the conversation, but they are not expected to actively contribute, although they can express themselves in a simple way in Kalamang ... | This is a Retrieval-Augmented Generation task for improving response quality in dialogue systems. Given a user query: {user query} and a set of retrieved documents: {retrieved documents}, the goal is to generate a coherent and contextually relevant response. Please generate a response that integrates information from the retrieved documents to accurately address the user's query. | D |
| **Method** | | **Output Results** | |
| GPT-4o mini | | A | |
| GWM-E | | D | |

*Table 31.* **Planning and optimization task and output comparison.** Agent decision-making prediction using ALFWorld dataset. Results show GWM-E vs T5 FT baseline.

| Task Description | | | |
|---|---|---|---|
| **Task** | | Planning and optimization | |
| **Image** | **Text** | **Action Prompt** | **Ground Truth** |
|  | Task: put a potato in countertop | Embodied household task: predict next decision-making behavior based on multimodal information. | go to garbagecan 1 |
| **Method** | | **Output Results** | |
| T5 FT | | go to microwave 1 | |
| GWM-E | | go to garbagecan 1 | |

