# OpenReview forum: "Graph World Model"
_ICML.cc/2025/Conference — ICML 2025 poster_

### Official Review · Reviewer_6SxJ · 2025-02-25

**Overall Recommendation:** 4

**Summary:**

This paper introduces the first world model in the graph domain. The proposed method is capable of handling multimodal graph data within a unified framework, accomplishing tasks across multiple domains, including prediction, generation, and optimization. The graph world model demonstrates exceptional performance and generalization capabilities on these tasks. Additional experiments highlight the role of graph structures in these tasks and the ability of the graph world model to leverage them effectively.

**Claims And Evidence:**

The claims in this paper are well validated, with no significant over-claiming.

**Essential References Not Discussed:**

I believe the paper does not have any significant references not discussed.

**Experimental Designs Or Analyses:**

The experiments are diverse and well-chosen, but there are some issues with task descriptions:

1) **Missing setails** Taking graph prediction as an example, the task splits and the comparison methods are not clearly described. For instance, in the case of supervised GCN, the settings should differ significantly from those of the Graph World Model. How GCN utilizes textual information and other details are not provided. In fact, in a supervised setting, the performance of GNNs should be better.

2) **Experiment costs** The cost of using the Graph World Model has also not been reported. As a reader more focused on graph-related tasks, I am concerned with the differences in training and inference efficiency between the Graph World Model and other foundation models, as well as the feasibility of deploying the Graph World Model to large-scale graphs.

3) **Examples** The content on world generation and world optimization needs large revisions.
First, for the generation task, examples of generated outputs should be provided for comparison.
Second, the input and output of the tasks should be clearly explained. I understand that the authors have simplified the descriptions in the main text due to space limitations, but the supplementary details in the appendix are still limited and not enough to help readers who are not deeply familiar with these tasks understand the task settings and configurations.

4) **A mInor issue** The distinction between world generation and world optimization is unclear to me. They seem to be essentially the same type of generation task. Adding examples, outputs, and further clarifications for each task would be very helpful, especially given the broad range of domains covered in this paper.

**Methods And Evaluation Criteria:**

The proposed approach makes sense, but its core idea still relies on mixing structural information into LLM and LVM, followed by fine-tuning techniques. The innovation lies in unifying different tasks, which is commendable, but the utilization of graph structures is somewhat limited.

**Other Comments Or Suggestions:**

N/A

**Other Strengths And Weaknesses:**

The strength of this paper lies in its exploration of an interesting task setting, and integrating the World Model is a great attempt at advancing graph foundation models. Additionally, the experiments are well-designed and cover a broad range of domains, making this paper one of the most comprehensive I have read on graph foundation models.

The main issues with the paper are clarity and technical novelty. In terms of clarity, the descriptions of the task details are too rough, and the comparisons between sub-tasks are limited, making it difficult to evaluate the performance of each task individually. This issue ties into the limited technical novelty that was mentioned earlier.

**Questions For Authors:**

The main questions have been listed before, and other issues require the authors to provide additional details. Once I have a clearer understanding of the experimental setup, I could raise further points.

Other questions:
1) Why is INSTRUCTG2I not included as baselines?
2) In [1], multi-modal information, including images and text, has been shown to benefit predictors in traditional graph tasks. Can the graph world model achieve that as well?

[1] Zhu, Jing, et al. "Multimodal graph benchmark." arXiv preprint arXiv:2406.16321 (2024).

**Relation To Broader Scientific Literature:**

I believe this paper has a good broader impact. The concept of the World Model is an intriguing research direction, and exploring its potential to unify graph tasks would be a valuable attempt. It also provides inspiration for future work in this area.

**Theoretical Claims:**

N/A

---

> ### Author Rebuttal · Authors · 2025-04-01
>
> **Response to Issue 1 (Missing details):** Thanks for the comments. The settings for the baselines primarily follow LLAGA (Chen et al., 2024a) and OFA (Liu et al., 2023a). As stated in Appendix A.3, we convert all nodes and labels in the Cora, PubMed, and HIV datasets into text. For all methods, we use BERT to obtain text embedding. We divide all datasets into training, validation, and test sets in an 8:1:1 ratio. We will add these details to the next version.
>
> **Response to Issue 2 (Experiment costs):** Thanks for your questions. We answer them one by one:
>
> ***[Training and inference efficiency.]*** Accurately comparing the training and inference efficiency of GWM with other FMs is highly challenging because many FMs are not designed to address multimodal problems and utilize various architectures. We can only compare the efficiency between GWM and LLM-based FMs from principles. For GWM-T, its efficiency shows no fundamental difference from other LLM-based FMs, as both are based on the standard instruction tuning. For GWM-E, its training process only requires fine-tuning the projector as stated in ***[lines 275-277]***, and its embedding-based method also saves a significant amount of token cost, making it more efficient. For GWM-E, it takes \~7 hours (\~$\frac{1}{4}$ of GWM-T) of training time on four NVIDIA A6000 GPUs described in ***[lines 352-376]***, and the inference time per case averages 0.213s (similar to GWM-T). Moreover, GWM-E significantly reduces memory usage with a shorter token length of 140.23 (~$\frac{1}{14}$ of GWM-T).
>
> ***[Large-scale graphs feasibility.]*** GWM-E can be deployed to large-scale graphs. In section 4.2, we designed a simplified GCN. It not only reduces computational complexity (Wu et al., 2019; He et al., 2020a) but also benefits from PyTorch's sparse matrix multiplication acceleration, allowing it to scale efficiently to large graphs. As shown in ***Tables 10 and 11***, we have conducted experiments with large graphs at the scale of hundreds of thousands, demonstrating GWM's ability to address computational complexity and scalability.
>
> **Response to Issue 3 (Examples) and 4 (A mInor issue):** Thanks for the feedback. These tasks can all be framed as generation tasks when using LLM to construct a GWM. But as stated in section 2.3, the essence of world optimization is to guide agents in decision-making, while world generation primarily enables agents to perform QA responses and generation, thus, their objectives are different. We distinguish these tasks to better demonstrate the capabilities of GWM in various aspects.  Additionally, we have provided detailed descriptions of each task in ***section 2.3, section 5, Table 9, and Tables 13-22*** of the paper. However, following the reviewers' suggestions, we have supplemented the details in Tables 1-7 of [rebuttal.pdf](https://anonymous.4open.science/r/ICML_2025_14809-7B6C/rebuttal.pdf). We will add these details to the next version.
>
> **Response to utilization of graph structures:**  Thanks for the comment.  Both GWM versions fully leverage graph structures for message passing from the perspectives of tokens and embeddings. Furthermore, we verify through the experiment in section 5.2 (No Graph vs N-hop) that graph structures enhance performance on downstream tasks.
>
> **Response to Question 1:** Thanks for your question. We didn't include INSTRUCTG2I (with complex QFormer design) as a baseline since our goal was to design a lightweight framework like GWM-E. Following your advice, we compared performance on multi-modal generation tasks as follows. Results show INSTRUCTG2I outperforms GWM on Goodreads but underperforms on Multi-Modal-Paper. This may be because the large-parameter INSTRUCTG2I also requires a larger dataset to achieve good results. In contrast, our lightweight GWM offers better efficiency while adapting well to multiple scenarios. We'll include this discussion in our next version.
>
> | Model | Goodreads |  | Multi-Modal-Paper |  |
> |-------|-----------|-----------|-------------------|-----------|
> |  | CLIP | DINOv2 | CLIP | DINOv2 |
> | INSTRUCTG2I | **50.37** | **25.54** | 56.37 | 18.80 |
> | GWM-T | 47.46 | 20.91 | **59.92** | 23.10 |
> | GWM-E | 45.23 | 20.87 | 59.84 | **26.03** |
>
> **Response to Question 2:** To validate whether similar conclusions to those in [1] can be achieved on GWM, we conducted an ablation study. Specifically, we designed baselines named Text-only, which means training and testing solely on the text modality compared with GWM-E. We made comparisons in planning and optimization (ALFWorld) as follows. It can be observed that removing other multimodal information leads to a performance decline in GWM, which aligns with the conclusions of [1]. We will conduct more ablation studies in the next version to further substantiate this finding.
>
> | Method | Precision | Recall | F1 |
> |--------|-----------|--------|-----|
> | Text-only | 92.95 | 92.80 | 92.86 |
> | GWM-E | **97.31** | **96.52** | **96.17** |

---

> > ### Comment · Reviewer_6SxJ · 2025-04-03
> >
> > Thanks to the authors' comprehensive feedback!
> >
> > Most of my concerns have been addressed, and I'd like to increase my score to 4.

---

> > > ### Author Response · Authors · 2025-04-04
> > >
> > > Thank you for your thoughtful and constructive feedback. We are pleased to hear that our responses have addressed most of your concerns. We are committed to incorporating the suggested changes in our revisions to further enhance the manuscript.

---

### Official Review · Reviewer_a6Z5 · 2025-03-14

**Overall Recommendation:** 3

**Summary:**

This paper proposes Graph World Model (GWM), a novel framework that integrates graph-structured data and multi-modal information into a unified world model for diverse tasks including prediction, generation, and planning. The authors present two variants (GWM-T and GWM-E) with distinct message-passing strategies and demonstrate competitive performance across six domains compared to domain-specific baselines. While the idea of unifying graph-based reasoning with multi-modal world modeling is promising, the technical depth and methodological clarity need further refinement to fully establish its novelty and scalability.

**Claims And Evidence:**

The claims and evidence make sense for the generalization problem.

**Essential References Not Discussed:**

The important references are fully discussed from my point of view.

**Experimental Designs Or Analyses:**

The experiments provide strong empirical validations for the proposed method. However, the analyses on the important hyperparameters are weak.

**Methods And Evaluation Criteria:**

The proposed method is reasonable. The evaluation criteria are appropriate for the problem.

**Other Comments Or Suggestions:**

I would suggest that the authors formalize the connection to world models and analyze trade-offs between graph complexity and task performance.

**Other Strengths And Weaknesses:**

The strengths of this paper are as follows:
(1) The integration of graph message-passing into multi-modal world models is a novel contribution, enabling unified handling of structured and unstructured data across prediction, generation, and optimization tasks.
(2) The empirical validation highlights that multi-hop aggregation consistently improves performance (e.g., in recommendation and RAG), underscoring the importance of graph structure for complex reasoning.
(3) The demonstrated zero-shot/few-shot capabilities on unseen tasks suggest practical value for low-resource scenarios.

The weaknesses of this paper are as follows:
(1)	Ambiguity in Multi-modal Fusion and Aggregation:
a)	While GWM-T converts multi-modal data into text tokens and GWM-E concatenates CLIP/BERT embeddings (Section 3.1, 4.1), the rationale for avoiding advanced fusion techniques (e.g., cross-modal attention) is unclear. For instance, tokenization in GWM-T may lose fine-grained visual semantics, and simple concatenation in GWM-E risks modality dominance (e.g., text overshadowing images).
b)	The parameter-free multi-hop aggregation (Section 4.2) adopts a "simplified GCN" design without adaptive mechanisms (e.g., attention in GATv2 or dynamic pruning in GRIT). This limits its ability to handle heterogeneous graphs or dynamically adjust aggregation weights during inference, potentially underutilizing graph structure (e.g., implicit vs. explicit edges in Section 2.2).
(2)	The paper positions GWM as a "world model" but lacks a formal connection to classical Markovian state transitions (Section 2.1). For example, how do graph-based actions (Section 2.2) align with traditional WM components like transition probability? The framework currently resembles a graph-enhanced multi-modal system rather than a principled extension of world models.
(3)	The simplified message-passing mechanism lacks theoretical grounding. For instance:
a)	How does the parameter-free design (Equation 4.2) avoid over-smoothing in deep aggregation?
b)	Does performance gain primarily stem from graph homophily (e.g., node similarity), and how does GWM handle non-homophilous graphs (common in real-world scenarios)?
c)	What guarantees exist for the alignment between graph transitions and task objectives (e.g., RAG retrieval vs. graph-level optimization)?

**Questions For Authors:**

(1)	Why prioritize parameter-free aggregation over learnable mechanisms (e.g., attention)? How does this design impact performance on heterogeneous graphs?
(2)	How are modality-specific embeddings aligned in GWM-E? Is there empirical evidence of modality dominance (e.g., text overshadowing images)?
(3)	Can the framework theoretically justify the relationship between graph transitions and task objectives (e.g., RAG retrieval vs. planning)?

**Relation To Broader Scientific Literature:**

The paper is well-situated within the literature on graph foundation model, multi-modal diffusion and graph-based retrieval-augmented generation

**Theoretical Claims:**

The detailed theoretical analyses are missing

---

> ### Author Rebuttal · Authors · 2025-04-01
>
> **Response to Weaknesses 1:**  Thanks for your questions. We answer your questions one by one:
>
> ***[Rationale of cross-modal fusion of GWM.]*** There are many ways to perform multimodal fusion. We selected two representative methods, not to avoid advanced fusion techniques. One is a simple and direct method that unifies multimodal information into text via GWM-T. The other method uses GWM-E, employing a cross-modal projector to unify different modal embeddings efficiently. Moreover, GWM-E does not have a modality dominance issue. We designed baselines named No-image and No-text, which mean training and testing solely on the text modality or image modality compared with GWM-E. We made comparisons on ALFWorld as follows. We can observe that the impact of images and text on GWM-E is almost equivalent, which demonstrates that GWM-E does not have modality dominance and all modalities are important. We will conduct more ablation studies in the next version to further substantiate this finding.
>
> | Method | Precision | Recall | F1 |
> |--------|-----------|--------|-----|
> | No-image | 92.95 | 92.80 | 92.86 |
> | No-text | 92.32 | 91.92 | 92.10 |
> | GWM-E | **97.31** | **96.52** | **96.17** |
>
> ***[Rationale of simplified GCN and its adaptiveness to heterogeneous graphs.]*** Indeed, our main experiment is based on homogeneous graphs. Because heterogeneous graphs often rely on predefined node and edge types, but we have multiple tasks and also need the model to generalize to unseen tasks, it is hard to define node and edge types in advance. If heterogeneous type information is represented as tokens or embeddings similar to GWM, ordinary GNNs may solve this, but it would lead to complexity in implementation and be costly. A potential way to extend directly from GWM-E to heterogeneous graphs is to perform separate multi-hop aggregations based on different types of edges and then flatten the resulting node embeddings into sequences to feed into the LLM decoder. However, we believe that our results on six real-world tasks validate GWM’s effectiveness and efficiency, highlighting our contribution. We will leave the exploration of heterogeneous graphs to future work.
>
> **Response to Weaknesses 2:** Thanks for your comments. We have already detailed how GWM models state transitions in section 2.2. As described in ***[lines 56-73]***, we use a graph for general multimodal state modeling. Moreover, to address the diversity of action settings in traditional WM, we model actions as action nodes (***[lines 75-88]***) and unify them into text descriptions as in ***[section 3.3 and section 4.3]***. This allows GWM to use a unified model to solve multi-tasks. Lastly, in ***[lines 89-93 of section 2.2]***, we explain that our GWM essentially learns a transition function, which is aligned with principled world models.
>
> **Response to Weaknesses 3:**  Thanks for your comments. We answer your questions step by step:
>
> ***[Avoid over-smoothing.]*** As introduced in ***[lines 247-253]*** of section 4.2, we input the multi-hop embeddings into the decoder. This is equivalent to a skip connection, a standard practice that alleviates oversmoothing [1, 2].
>
> ***[GWM handles non-homophilous graphs.]*** Essentially, both versions of GWM transform graphs into a standard form interpretable by LLMs. Existing work (Liu et al., 2023a; Chen et al., 2024a) has demonstrated that LLMs can also effectively understand graphs, thus we anticipate that LLMs may outperform GNNs in comprehending homophilous or non-homophilous graphs. But if we must understand homophilous graphs from the perspective of GNNs, a potential way is to integrate [3] based on GWM-E, but this would introduce complex implementation and significant costs. Moreover, the performance across six representative real-world tasks has already validated the practicability and generalizability of GWM. We will include the above discussions in the next version.
>
> ***[Graph transitions align with task objectives.]***  As discussed in section 2.3, our action is essentially a description of the task objective. For example, in the RAG task, the action of GWM is to answer specific queries with retrieved contexts. Thus, the graph transition modelled by GWM is aligned with the task objective without a theoretical gap.
>
> **[1]** Optimization of graph neural networks: Implicit acceleration by skip connections and more depth.
>
> **[2]** Representation learning on graphs with jumping knowledge networks.
>
> **[3]** Revisiting heterophily for graph neural networks.
>
> **Response to missing theory and hyperparameters analysis:**  Thanks for the comment. We prioritize empirical analysis of GWM across multiple tasks over theoretical proofs, which are challenging in LLM research. Given the foundation model's numerous hyperparameters and computational costs, we focused on introducing key hyperparameters (lines 357-372 and Appendix B) and analyzed critical ones like hop num (section 5.2), balancing reproducibility with practical costs.

---

> > ### Comment · Reviewer_a6Z5 · 2025-04-07
> >
> > Thanks for the response. Most of my concerns have been addressed. I will raise the score to weakly accept.

---

> > > ### Author Response · Authors · 2025-04-07
> > >
> > > Thank you for your thoughtful and constructive feedback. We are glad to know that our responses have addressed most of your concerns. We will carefully incorporate your suggestions in our next version to further improve the quality of the manuscript.

---

### Official Review · Reviewer_4jVH · 2025-03-17

**Overall Recommendation:** 3

**Summary:**

In this paper, the authors propose Graph World Model (GWM), a framework designed to integrate both unstructured and graph-structured data with multi-modal information. They introduce two GWM variants: a token-based method that transforms multi-modal data into textual representations prior to message passing, and an embedding-based method that operates within a unified embedding space. The framework further incorporates action nodes to address diverse tasks and applies message-passing algorithms to combine structured information. The authors demonstrate competitive or superior performance against domain-specific baselines across six tasks and multiple domains. However, limited experimental comparisons, subpar presentation, and insufficient analysis negatively impact the paper’s overall quality.

**Claims And Evidence:**

The authors propose a framework that is clear.

**Essential References Not Discussed:**

n/a

**Experimental Designs Or Analyses:**

1. **Limited experimental comparison**. The experiments lack comprehensiveness in several ways. (1) Table 6 fails to clarify whether baselines were fine-tuned on task-specific datasets, making it difficult to assess the true comparative performance. (2) The authors should have included baseline graph world models (Zhang et al. 2021; Zhu et al. ICLR 2023) in their experiments, particularly for world planning and optimization tasks, to properly demonstrate the advantages of their proposed methods.
2. **Scalability and efficiency analysis**. The paper fails to address computational complexity and scalability considerations adequately. For real-world applications involving large graphs, multi-hop message passing could become computationally prohibitive, yet this limitation receives insufficient attention.

3. **Lack of qualitative analysis**. The authors present only quantitative results without providing qualitative comparisons, particularly for world generation tasks. Including actual examples of model outputs would help readers better understand the practical differences between GWM and baselines.

**Methods And Evaluation Criteria:**

- The paper bridges world models with graph-based approaches, addressing a clear gap in the literature. This integration allows handling both structured and unstructured data in a unified framework.

- The benchmark datasets makes sense. The experimental evaluation across six diverse tasks (multi-modal generation/matching, recommendation, graph prediction, multi-agent collaboration, RAG, and planning/optimization) demonstrates the versatility of the approach.

**Other Comments Or Suggestions:**

To enhance the presentation quality, the authors are encouraged to refine the figures and incorporate visualized examples to better illustrate the pipelines.

**Other Strengths And Weaknesses:**

**Other Weaknesses**

**Poor presentation affecting readability**. Figure 3, which illustrates the GWM framework, suffers from quality issues that impede understanding. (1) Target nodes are not clearly specified—variables should be directly labeled. Additionally, the target nodes and prompt in the top-right of Figure 3 are two separate inputs but are not represented with distinct arrows. (2) The use of Stable Diffusion in the top-right creates confusion since it was designed for text-to-image generation, while the diagram shows token inputs and "next state" outputs. The authors should clarify that the model operates in Stable Diffusion's latent space.

**Questions For Authors:**

Most of the baselines are relatively weak (e.g., Table 6). Can the author supplement the baseline that tuned on the task-specific dataset.

**Relation To Broader Scientific Literature:**

The paper inadequately differentiates its methodology from previous graph world models such as Zhang et al. The authors focus primarily on application differences in the related work section rather than providing a thorough methodological comparison, which would better highlight their novel contributions.

**Theoretical Claims:**

This paper mainly provides the empirical analysis.

---

> ### Author Rebuttal · Authors · 2025-04-01
>
> **Q1. Can the author supplement the baseline that tuned on the task-specific dataset (e.g., Table 6).**
>
> **Response:** Thanks for the comments. Indeed, we have detailed the specific settings of the baselines for Table 6 in Appendix A.4 (***[lines 779-796]***). We primarily selected three classic agent baselines for comparison: CoT, ToT, and Few-shots. Although these baselines were not additionally fine-tuned on this dataset, the results of GWM shown in Figure 6, as described in ***[lines 382-384]***, were also not additionally fine-tuned. Therefore, from this perspective, the comparison between GWM and the baselines is fair. However, following the reviewer's suggestion, we compared two baselines fine-tuned on AgentClinic: FT is a baseline finetuned on the task using the Llama-3-8B model. Longformer [1] is a strong baseline for long document understanding. The table below reports the comparison results between GWM and these baselines, demonstrating the effectiveness of GWM in AgentClinic.
>
> | Method     | Acc       | Recall    | F1        |
> |------------|-----------|-----------|-----------|
> | FT         | 45.00      | 45.40     | 44.00     |
> | Longformer | 25.00      | 20.20     | 14.00     |
> | GWM-T      | **50.00** | **46.42** | **48.20** |
> | GWM-E      | 45.00     | 39.57     | 35.56     |
>
> ***[1]*** Longformer: The long-document transformer.
>
>
> **Q2. The authors should have included baseline graph world models (Zhang et al. 2021; Zhu et al. ICLR 2023) in their experiments.**
>
> **Response:**  Thanks for your feedback. Indeed, the works on graph world models (Zhang et al. 2021; Zhu et al. ICLR 2023) do not consider the processing of multimodal information in their modeling and primarily focus on offline scenarios in the RL field. These limitations restrict their capabilities as world models and also make them inapplicable for direct comparison in our scenario. In contrast, GWM is a world model capable of handling multimodal information and generalizing across various tasks. We will supplement these discussions in the related work section of the next version.
>
> **Q3. The paper fails to address computational complexity and scalability considerations adequately.**
>
> **Response:** Thanks for your valuable comments. In fact, we address computational complexity and scalability from two perspectives. First, we introduce GWM-E, which, as discussed in ***[lines 275-277]***, fixes the LLM's parameters and only fine-tunes the projector parameters, thereby reducing computational complexity. Additionally, as described in section 4.2 ***[lines 234-246]***, we designed a simplified, parameter-free GCN. This not only reduces computational complexity compared to traditional GNNs (Wu et al., 2019; He et al., 2020a) but also benefits from PyTorch's sparse matrix multiplication acceleration, allowing it to scale more effectively to large graphs. As shown in ***Tables 10 and 11*** in the appendix, we have conducted experiments with large graphs at the scale of hundreds of thousands, demonstrating GWM's ability to address computational complexity and scalability.
>
> **Q4. The authors present only quantitative results without providing qualitative comparisons.**
>
> **Response:**  Thanks for the insightful advice. We have already added qualitative comparisons for all tasks in  Tables 1-7 of [rebuttal.pdf](https://anonymous.4open.science/r/ICML_2025_14809-7B6C/rebuttal.pdf). We will complete this information in the appendix of the next version.
>
> **Q5. Figure 3, which illustrates the GWM framework, suffers from quality issues that impede understanding. (1) Target nodes are not clearly specified—variables should be directly labeled. Additionally, the target nodes and prompt in the top-right of Figure 3 are two separate inputs but are not represented with distinct arrows. (2) The use of Stable Diffusion in the top-right creates confusion since it was designed for text-to-image generation, while the diagram shows token inputs and "next state" outputs. The authors should clarify that the model operates in Stable Diffusion's latent space.**
>
> **Response:** Thanks for your valuable suggestions. Following your suggestions, we first specified the target nodes in Figure 3. Next, regarding the issue with the top right arrow, we intended to prompt the target nodes, so we changed the content on the arrow to 'Prompt (Target nodes)'. Lastly, concerning the input and output issues of Stable Diffusion in GWM-T, as described in ***[lines 229-234]***, our input included action nodes and target nodes at the token level, and the output image served as the next state. To further clarify this process, we added legends to distinguish between different colors and shapes. We have updated the modified figure in Figure 2 of [rebuttal.pdf](https://anonymous.4open.science/r/ICML_2025_14809-7B6C/rebuttal.pdf) and it will be updated in the next version.

---

### Official Review · Reviewer_5yBi · 2025-03-24

**Overall Recommendation:** 4

**Summary:**

This paper proposes a Graph World Model that supports both unstructured and graph-structured states with multi-modal data. The proposed model can tackle diverse sets of tasks and act as a graph-based foundation model. The results on numerous datasets and tasks show SOTA or comparable results on most tasks compared to the baselines.

**Claims And Evidence:**

- The paper claims to propose a graph world model that can tackle multidomain tasks and be applied to both structured and unstructured
data.
  - The claim is backed by numerous experiments and empirical results on multidomain and multimodal datasets with different combinations of structured (graphs) and unstructured (text, image).
- The claims are clear and the results are convincing.

**Essential References Not Discussed:**

Although the proposed method in [a] is different from the one here, they could potentially solve similar tasks. It would be nice to contrast or compare against it.

[a] Ebrahimi, Sayna, et al. "LANISTR: Multimodal Learning from Structured and Unstructured Data." CoRR (2023).

**Experimental Designs Or Analyses:**

The designed experiments are sound and valid and show the analysis of the proposed model in different tasks, datasets, and scenarios.

**Methods And Evaluation Criteria:**

Yes, the proposed method claims to act as a foundation model, and the results on 11 datasets compared to the baselines on those datasets show better or comparable results.

**Other Comments Or Suggestions:**

1. The graphics can be improved to make the method clearer and more understandable. Eg. in Figs. 1,3 it is not clear what each color identifies in Fig. 1: what is the difference between intended and unintended action, and what is the difference between the node/edge/graph level actions?

**Other Strengths And Weaknesses:**

Strengths:
1. The paper is very well-written, and it was a delight to read.
2. The contributions are clear, and the claims are backed by the experiments.
3. The model is extensively evaluated on multiple tasks and datasets and shows SOTA / comparable results in most scenarios.

Weaknesses:
1. It would be nice to include the SOTA results for the different tasks. Although for most tasks, the baselines are recent, in Tab. 4, the provided baselines are old. Maybe the authors can include FREEDOM [e] or another recent SOTA model for recommendation.


[e] Zhou, Xin, and Zhiqi Shen. "A tale of two graphs: Freezing and denoising graph structures for multimodal recommendation." Proceedings of the 31st ACM International Conference on Multimedia. 2023., https://dl.acm.org/doi/pdf/10.1145/3581783.3611943

**Questions For Authors:**

Could the authors contrast their work against LANISTR [a] for general multimodal learning and, if possible, compare to FREEDOM [e]?

**Relation To Broader Scientific Literature:**

The paper seems to be novel to the best of my knowledge in terms of using a graph-based world model that considers states and actions as graphs and enables the use and combination of multimodal structured and unstructured data, which is valuable. The method, although different in nature, solves a similar task to LANISTR [a], which would be nice to cite and contrast against.

**Theoretical Claims:**

N/A

---

> ### Author Rebuttal · Authors · 2025-04-01
>
> **Q1. The method, although different in nature, solves a similar task to LANISTR [a], which would be nice to cite and contrast against.**
>
> **Response:**  Thanks for the valuable suggestions. Indeed, we have already compared two baselines that, like LANISTR, were pre-trained by modality alignment and then fine-tuned on downstream tasks, Contrastive MLP (Liu et al., 2022), CLIP FT (Radford et al., 2021) in ***[lines 348-352] and [Table 3]***. However, we followed the reviewer's suggestions and compared the performance of GWM with LANISTR. Due to the limited time for rebuttal, we primarily focused on comparing the performance of GWM and LANISTR when directly finetuned on the Goodreads dataset in the task of multi-modal matching. It can be observed that GWM achieved a significant performance gain compared to LANISTR. In the next version, we will compare LANISTR in more tasks and evaluate its performance after pretraining on our dataset.
>
> | Model | Accuracy | Recall | F1 Score |
> |-------|-----------|---------|----------|
> | LANISTR | 54.3 | 51.8 | 43.6 |
> | GWM-T | 84.22 | 85.66 | 85.29 |
> | GWM-E | **88.82** | **89.73** | **89.06** |
>
> **Q2. In Tab. 4, the provided baselines are old. Maybe the authors can include FREEDOM [e] or another recent SOTA model for recommendation.**
>
> **Response:**  Thank you for your insightful comments. In the recommendation task, we primarily selected some classic and representative baselines for comparison. Our goal in comparison with task-specific baselines is not to demonstrate that GWM can beat sota in all tasks, but to show that GWM, with a unified model, has good generalization capabilities across many tasks, including recommendation. However, we followed the reviewer's suggestions and compared the performance of GWM with FREEDOM on the recommendation task shown in the following table. We can observe that, compared to FREEDOM, the two methods of GWM achieved better performance, which demonstrates the effectiveness of GWM. We will update the experimental results and discussion in the next version.
>
> | Model | Baby |  | Sports |  | Clothing |  |
> |-------|------|---------|--------|---------|----------|---------|
> | | Recall | F1 Score | Recall | F1 Score | Recall | F1 Score |
> | FREEDOM | 60.35 | 66.16 | 63.47 | 70.53 | 70.20 | 78.40 |
> | GWM-T | 70.84 | 75.08 | 84.29 | 88.60 | 71.73 | 74.26 |
> | GWM-E | **76.72** | **84.74** | **88.78** | **90.32** | **75.27** | **84.06** |
>
> **Q3. The graphics can be improved to make the method clearer and more understandable. Eg. in Figs. 1,3 it is not clear what each color identifies in Fig. 1: what is the difference between intended and unintended action, and what is the difference between the node/edge/graph level actions?**
>
> **Response:** Thanks for your valuable advice. Following your suggestions, we have separated the descriptions of different action levels and the methods for obtaining target nodes in Figure 1. Additionally, we have standardized the colors in corresponding parts of Figures 1 and 3. Lastly, we have added captions to each figure to differentiate the meanings of various colors and shapes. We showed the figures in Figure 1 and 2 of [rebuttal.pdf](https://anonymous.4open.science/r/ICML_2025_14809-7B6C/rebuttal.pdf). We will update these modifications in the next version.

---

### Decision · Program_Chairs · 2025-05-01

**Decision:**

Accept (poster)

**Comment:**

This paper investigates the challenge of addressing both unstructured and graph-structured states with multi-modal information, representing diverse tasks as actions. To tackle this, the authors propose the Graph World Model (GWM), which leverages a generic message-passing algorithm to aggregate structured information. This can be done either over a unified multi-modal token space by converting multi-modal data into text (GWM-T) or a unified multi-modal embedding space using modality-specific encoders (GWM-E). Experiments on several datasets demonstrate the effectiveness of the proposed model.


Strengths:
1. The paper is well-written, and easy to follow.
2. The paper successfully bridges world models with graph-based approaches, addressing a clear gap in the literature. This integration enables the unified handling of both graph-structured and unstructured data.


Weaknesses:
1. Some experiments lack sufficient baselines to convincingly demonstrate the effectiveness of the proposed model. Additional aspects of the experiments, such as scalability and efficiency analysis, as well as a more thorough qualitative analysis, should be enhanced to make the results more persuasive.
2. The theoretical analysis could be expanded to provide more depth and clarity.




Overall, I recommend that the authors further refine the paper in terms of experiments and theoretical analysis to ensure a more comprehensive and convincing presentation in the camera-ready version.